# EGF receptor signaling, phosphorylation, ubiquitylation and endocytosis in tumors in vivo

Itziar Pinilla-Macua[1], Alexandre Grassart[2], Umamaheswar Duvvuri[3], Simon C Watkins[1], Alexander Sorkin[1]*

[1]Department of Cell Biology, University of Pittsburgh School of Medicine, Pittsburgh, United States; [2]Department of Molecular Microbial Pathogenesis, Institute Pasteur, Paris, France; [3]Department of Otolaryngology, University of Pittsburgh School of Medicine, Pittsburgh, United States

**Abstract** Despite a well-established role for the epidermal growth factor receptor (EGFR) in tumorigenesis, EGFR activities and endocytosis in tumors in vivo have not been studied. We labeled endogenous EGFR with GFP by genome-editing of human oral squamous cell carcinoma cells, which were used to examine EGFR-GFP behavior in mouse tumor xenografts in vivo. Intravital multiphoton imaging, confocal imaging of cryosections and biochemical analysis revealed that localization and trafficking patterns, as well as levels of phosphorylation and ubiquitylation of EGFR in tumors in vivo closely resemble patterns and levels observed in the same cells treated with 20–200 pM EGF in vitro. Consistent with the prediction of low ligand concentrations in tumors, EGFR endocytosis was kinase-dependent and blocked by inhibitors of clathrin-mediated internalization; and EGFR activity was insensitive to Cbl overexpression. Collectively, our data suggest that a small pool of active EGFRs is sufficient to drive tumorigenesis by signaling primarily through the Ras-MAPK pathway.

DOI: https://doi.org/10.7554/eLife.31993.001

*For correspondence:
sorkin@pitt.edu

**Competing interests:** The authors declare that no competing interests exist.

## Introduction

For the last four decades, EGFR, the receptor for epidermal growth factor (EGF), has been the predominant experimental model for studying the family of receptor tyrosine kinases. EGFR is an essential player during mammalian development and involved in the maintenance of various tissues in adult organisms (*Sibilia et al., 2007*; *Pastore et al., 2008*). Because EGFR is mutated or overexpressed in a variety of human cancers, it has become a major prognostic marker and therapeutic target (*Grandis and Sok, 2004*; *Herbst et al., 2004*; *Nicholson et al., 2001*). Pharmacologic inhibition of EGFR has been a successful strategy for the treatment of patients with non-small-cell lung carcinoma (NSCLC) expressing mutant EGFR (*Lee, 2017*). However, therapies targeting wild-type EGFR in tumors such as head and neck squamous cell carcinoma (HNSCC) have been less effective due to intrinsic and acquired tumor resistance to EGFR inhibitors (*Leemans et al., 2011*; *Rieke et al., 2016*; *Jiang et al., 2014*).

EGFR may be activated by seven different ligands that bind the extracellular domain of the receptor with varying affinity (*Singh et al., 2016*; *Roepstorff et al., 2009*; *Ebner and Derynck, 1991*). These ligands are synthetized as transmembrane precursors that undergo proteolysis by metalloproteases at the cell surface to produce functional soluble molecules (*Singh et al., 2016*). Ectodomain 'shedding' is proposed to be the rate-limiting step of the molecular processing that determines the concentration of a mature ligand in vivo (*Peschon et al., 1998*). Ligand binding leads to EGFR dimerization and the activation of its tyrosine kinase (*Lemmon and Schlessinger, 2010*). Subsequent

phosphorylation of tyrosine residues in the carboxyl-terminus of EGFR provides docking sites for proteins with SH2 and PTB domains, which trigger signal transduction through Ras-Raf-mitogen-activated protein kinase/extracellular-signal regulated kinase 1/2 (MAPK/ERK1/2), phosphoinositide 3-kinase, AKT, Src family kinases (SFKs), STATs, phospholipase Cγ1, Rho family GTPases and other pathways (*Lemmon and Schlessinger, 2010*).

Once ligand-bound, EGFR is rapidly endocytosed via clathrin-dependent and -independent mechanisms (*Sorkin and Goh, 2009*). Subsequently, EGFR-ligand complexes may recycle back to the plasma membrane from early and sorting endosomes but are also efficiently targeted to lysosomes for degradation (*Sorkin and Goh, 2009*). Recruitment of E3 ubiquitin ligases, the Cbl proteins (c-Cbl and Cbl-b), to activated EGFR is a key event leading to receptor ubiquitylation (*Levkowitz et al., 1998*; *Levkowitz et al., 1999*). Ubiquitylation of the receptor and its binding to the clathrin adaptor AP-2 serve as two redundant mechanisms for receptor recruitment into clathrin-coated pits (*Fortian et al., 2015*). Poly-ubiquitylation also mediates binding of EGFR to ESCRT proteins in sorting endosomes leading to the incorporation of the receptor into intraluminal vesicles of multi-vesicular endosomes and subsequent lysosomal degradation (*Eden et al., 2012*; *Huang et al., 2013*). Despite the extensive literature on EGFR endocytosis and intracellular sorting, the mechanisms underlying these processes and how they regulate EGFR signaling in normal and tumor cells are poorly understood.

A significant reason for the gaps in understanding of the physiologic mechanisms and function of EGFR endocytosis is the unusual threshold dependence of EGFR endocytosis and signaling on ligand concentration. Activation of <10,000 EGFRs per cell with low ligand concentrations (<1–2 ng/ml) results in EGFR internalization specifically via the clathrin-mediated endocytosis (CME) pathway, whereas at higher EGF concentrations the process becomes largely clathrin independent (CIE), presumably, due to saturation of the CME pathway (*Lund et al., 1990*; *Jiang and Sorkin, 2003*; *Sigismund et al., 2005*). Furthermore, a ligand concentration threshold was observed in EGFR ubiquitylation (*Sigismund et al., 2008*; *Sigismund et al., 2013*). It has been proposed that at low EGF concentrations (1–2 ng/ml) activated EGFRs are not ubiquitylated in HeLa cells, and therefore, not efficiently targeted to lysosomes, which leads to continued recycling and signaling (*Sigismund et al., 2008*; *Sigismund et al., 2013*). Importantly though, previous studies in B82 mouse fibroblasts have shown that endosomal sorting of EGFR is rapid in the presence of low EGF concentrations but is saturated by high ligand concentrations (*French et al., 1994*). Furthermore, a small pool of ligand-occupied EGFRs was shown to be sufficient to fully activate the ERK1/2 signaling pathway in several types of cultured cells, in contrast to other downstream signaling pathways which demand high EGFR ligand concentrations before significant activation is apparent (*Albeck et al., 2013*; *Shi et al., 2016*; *Krall et al., 2011*). The concentration of EGFR ligands in human body fluids and tumors is typically below 1–2 ng/ml, with the exception of breast milk, saliva and urine, fluids where EGF is more abundant though inaccessible to EGFRs (*Ishikawa et al., 2005*; *Rich et al., 2017*; *Dvorak, 2010*; *Connolly and Rose, 1988*; *Murdoch-Kinch et al., 2011*; *Hirata and Orth, 1979*; *Oka and Orth, 1983*). However, even given this knowledge, majority of studies of EGFR endocytosis and signaling utilize extremely high, non-physiological concentrations of EGFR ligands, for example, 100 ng/ml.

The second major reason for our limited understanding of the EGFR physiology and pathophysiology is the lack of quantitative, high-resolution studies of endocytosis and signaling of endogenous EGFR in vivo, particularly, in tumor models. What are the levels of EGFR activity in tumor cells in vivo? Is EGFR endocytosed in tumors in vivo in the presence of endogenous ligands? Is EGFR ubiquitylated in tumors? What are the mechanisms of EGFR endocytic trafficking in vivo in tumor cells? In the present study, we have begun to address these questions using mouse tumor xenografts of human HNSCC cells engineered using gene-editing to express GFP-tagged endogenous EGFR (EGFR-GFP). A combination of intravital imaging of tumors, high-resolution fluorescence microscopy of tumor sections and biochemical analysis demonstrated activation, ubiquitylation and endocytosis of EGFR in tumors in vivo. We found that an extremely small pool of EGFRs is activated and ubiquitylated in tumor xenografts, indicative of low (picomolar) concentrations of endogenous EGFR ligands accessible to receptors in these tumors. These few active EGFRs are capable of efficient endocytosis in a receptor-kinase-activity-dependent manner and are sufficient to drive tumorigenesis.

## Results

### Generation and characterization of gene-edited HSC3 cells expressing endogenous GFP-tagged EGFR

To study EGFR endocytosis and signaling in tumors, we have used the human oral squamous carcinoma HSC3 cell line as the primary experimental model (*Momose et al., 1989*). These HNSCC cells have five copies of the wild-type *EGFR* gene (canSar v3.0) and thus express ~$5 \times 10^5$ EGFRs per cell, which is 5–10-fold higher than EGFR levels in normal keratinocytes and fibroblasts. HSC3 cells produce tumors in athymic nude mice (*Momose et al., 1989*; *Kudo et al., 2003*), and the growth of HSC3 tumor xenografts is inhibited by blocking EGFR activity, indicating that these tumors are EGFR-dependent (*Kudo et al., 2003*; *Shintani et al., 2003*). Because overexpression of EGFR is observed in the majority of human HNSCC (*Leemans et al., 2011*; *Rieke et al., 2016*; *Grandis and Tweardy, 1992*), HSC3 cells is considered to be a suitable model to recapitulate human EGFR-dependent head-and-neck carcinoma.

To enable direct visualization of endogenous EGFR in tumor cells in vivo, EGFR was tagged with eGFP in HSC3 cells using a zinc-finger nuclease (ZFN)-based genome-editing method (*Doyon et al., 2011*) (*Figure 1A*). After two cycles of gene-editing and multiple rounds of clonal selection, several clonal pools of HSC3 cells (single HSC3 cells do not survive) were obtained, in which EGFR-GFP constituted 40–50% of total cellular EGFR protein (*Figure 1B–D*), indicating that 2–3 copies of *EGFR* gene were edited. Clonal pool B7F8 (further referred as HSC3/EGFR-GFP cells; *Figure 1B*) was selected for subsequent experiments based on the homogeneity of subcellular distribution of EGFR-GFP within the cell population and the similarity of cell morphology with that of the parental cells.

The dose dependency of EGFR phosphorylation at Tyr1068 and EGFR ubiquitylation on EGF concentration was essentially the same between HSC3/EGFR-GFP and the parental HSC3 cells (*Figure 1C–D*). When HSC3/EGFR-GFP cells were stimulated with EGF-Rhodamine (EGF-Rh), efficient endocytosis of EGF-Rh:EGFR-GFP complexes was observed in living cells as evidenced by the accumulation of 80–90% of these complexes in endosomes with only a minimal EGF-Rh presence at the cell surface after 12 min of continuous endocytosis (*Figure 1E*). Subcutaneous (s.q.) grafting of HSC3/EGFR-GFP cells into the flanks of athymic nude mice led to tumor formation (*Figure 1F*). Treatment of mice harboring HSC3/EGFR-GFP tumor xenografts with gefitinib, a small-molecule EGFR tyrosine kinase inhibitor, substantially reduced tumor growth, demonstrating that HSC3/EGFR-GFP tumors require EGFR tyrosine kinase activity to sustain tumorigenesis (*Figure 1F*). Together, these data confirm that the GFP tag does not affect EGFR function, and validate HSC3/EGFR-GFP cells as an appropriate experimental system to study EGFR signaling and trafficking in EGFR-dependent tumors in vivo.

### EGFR-GFP localization and trafficking in HSC3/EGFR-GFP tumor xenografts

To examine the localization dynamics of EGFR-GFP in living tumors, intravital imaging of HSC3/EGFR-GFP flank xenografts was performed using a multi-photon microscope as described in 'Materials and methods'. Time-lapse images were acquired up to 150 μm deep into the tumor. The bulk of EGFR-GFP was found to be located at the plasma membrane (*Figure 2A–B*). Clusters and/or vesicles containing EGFR-GFP, as well as their movement, appearance and disappearance, were occasionally observed in cells located at the periphery of tumor nodules (*Figure 2A–B*; *Figure 2—videos 1* and *2*). Of note, the GFP-positive puncta did not display significant motility in vivo, in contrast to thhe rapid movement of endosomes typically observed in cultured cells.

To systematically analyze the localization of EGFR-GFP in the entire tumor volume, GFP fluorescence was imaged on tumor cryosections using confocal microscopy. Large-area montage images of the tumor section demonstrate a predominantly plasma membrane localization for EGFR-GFP (*Figure 2C*). Intracellular GFP puncta were detected in a limited fraction of cells in the tumor (insets in *Figure 2C*). Consistent with the observations from intravital imaging, clusters or vesicles of EGFR-GFP were often seen in close proximity to the plasma membrane or could not be resolved from the plasma membrane even by the highest resolution confocal imaging. Many GFP-positive clusters/vesicles, including those that are proximal to the plasma membrane, contained early endosome antigen 1 (EEA.1), suggesting that this pool of EGFR-GFP is located in early/sorting endosomes

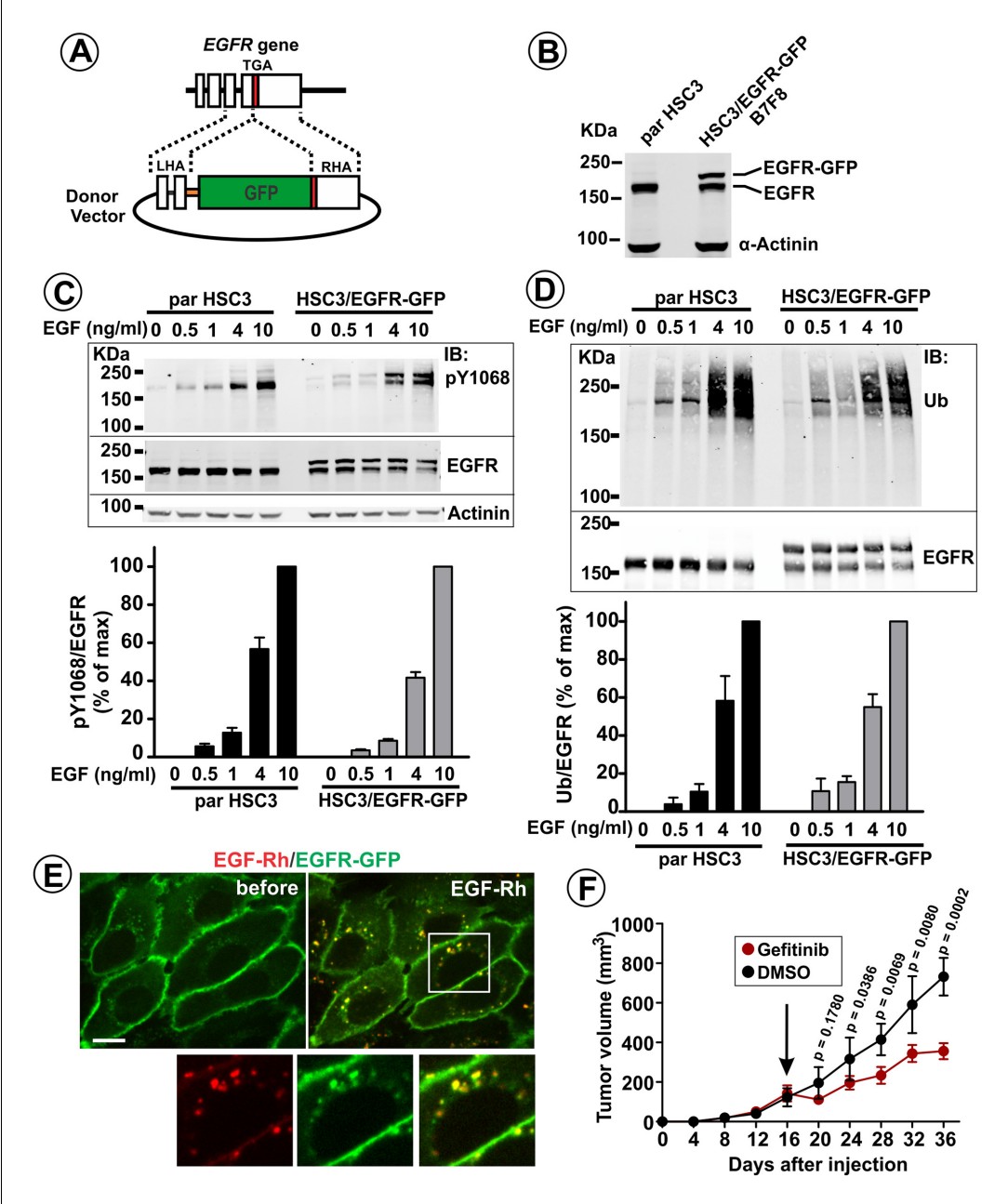

**Figure 1.** Generation and characterization of HSC3 cells expressing endogenous GFP-tagged EGFR. (**A**) Schematics of genome-editing. GFP sequence was inserted in-frame at the 3′-end of the coding sequence of the *EGFR* gene using a ZFN pair and a donor vector containing GFP inserted between left and right homology arms (LHA and RHA) from the genomic *EGFR* sequence. (**B**) Western blotting of parental (par) HSC3 and HSC3/EGFR-GFP cells (B7F8 clone) with the EGFR and α-actinin (loading control) antibodies. (**C**) Parental (par) HSC3 and HSC3/EGFR-GFP cells were stimulated with EGF for 10 min at 37°C and lysed. The lysates were probed by western blotting using antibodies to pY1068, EGFR and α-actinin (loading control). Bar graph represents mean values of ratios of pY1068 to total EGFR signals expressed as percent of the maximum value of the ratio at 10 ng/ml EGF (±S.E.M; n = 3). (**D**) Cells were stimulated with EGF for 10 min at 37°C and lysed. EGFR was immunoprecipitated, and the immunoprecipitates were probed by western blotting with ubiquitin and EGFR antibodies. Bar graph represents mean values of ratios of the amount of ubiquitylated EGFR to total EGFR expressed as percent of the maximum value of the ratio at 10 ng/ml EGF (±S.E.M; n = 3). (**E**) Live-cell imaging of HSC3/EGFR-GFP cells was performed through 488 nm (EGFR-GFP) and 561 nm (EGF-Rh) channels during stimulation of cells with 4 ng/ml EGF-Rh at 37°C. Merged images of individual frames before and 12 min after EGF-Rh stimulation are shown. Insets represent high magnification images of the region indicated by white rectangle. Scale bar, 10 µm. (**F**) HSC3/EGFR-GFP cells were implanted into flanks of athymic nude mice. Mice harboring tumors were randomized into two groups, which were administered with Gefitinib (30 mg/Kg) or vehicle (DMSO) i.p. 5 days/week for 3 weeks starting on day 16 when tumors reached ~100 mm³ (arrow). Averaged tumor volumes (±S.E.M; n = 6) are presented. Unpaired T-test was performed. p-Values < 0.05 are considered statistically significant.
DOI: https://doi.org/10.7554/eLife.31993.002

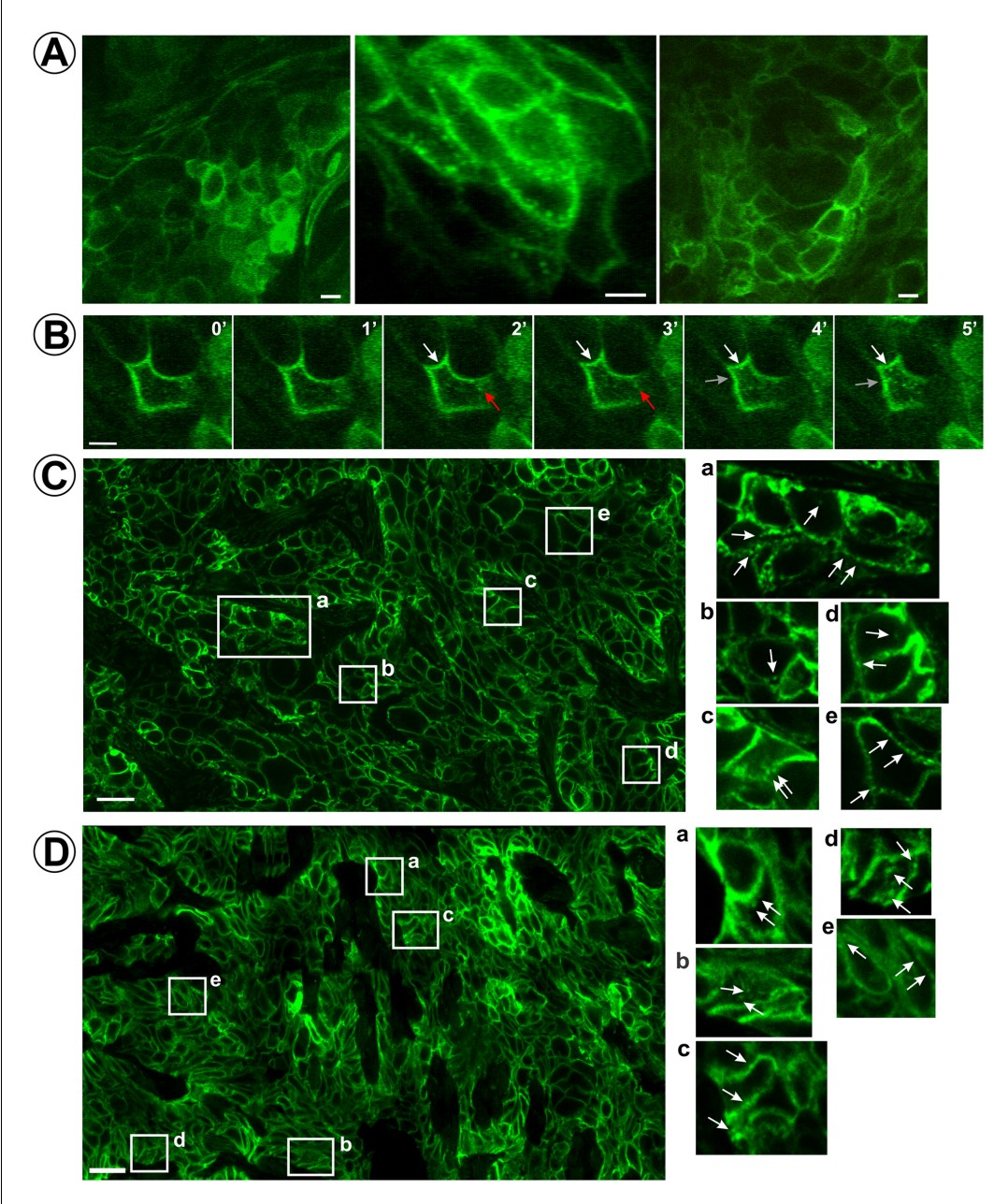

**Figure 2.** Localization and dynamics of EGFR-GFP in HSC3/EGFR-GFP tumor xenografts. (A and B) Time-lapse intravital imaging of HSC3/EGFR-GFP flank tumors was performed by multiphoton microscopy as described in 'Materials and methods'. (A) Representative snapshots from time-lapse series. See *Figure 2—video 1*. (B) Selected time-lapse images (0–5 min) of a single cell show the apparent formation of vesicles from the plasma membrane (white and red arrows). See *Figure 2—video 2*. Scale bars, 10 μm. (C and D) HSC3/EGFR-GFP flank (C) or tongue (D) tumors were dissected and fixed in paraformaldehyde. Confocal imaging of cryosections was performed through the 488 nm channel. Montage images of representative areas of tumors are shown. Insets represent high-magnification images of the regions indicated by white rectangles. Arrows point on the examples of clusters and vesicles of EGFR-GFP. Scale bars, 25 μm.

DOI: https://doi.org/10.7554/eLife.31993.003

The following video and figure supplement are available for figure 2:

**Figure supplement 1.** Immunofluorescence labeling of EGFR in HSC3/EGFR-GFP tumor flank xenografts.

DOI: https://doi.org/10.7554/eLife.31993.004

**Figure 2—video 1.** Time lapse imaging of HSC3/EGFR-GFP flank tumor xenograft.

DOI: https://doi.org/10.7554/eLife.31993.005

**Figure 2—video 2.** Time lapse imaging of HSC3/EGFR-GFP flank tumor xenograft.
*Figure 2 continued on next page*

*Figure 2 continued*

DOI: https://doi.org/10.7554/eLife.31993.006

(*Figure 3A*). The predominantly peripheral location of endosomes and their low motility is considered to be due to dense packing of cells in tumor nodules limiting the cytoplasmic volume in these cells as can be seen in *Figure 2—figure supplement 1*. Staining of tumor sections with EGFR antibodies demonstrated a localization pattern for the total EGFR (untagged and GFP-tagged EGFR) that is essentially identical to that seen for EGFR-GFP (*Figure 2—figure supplement 1*), thus confirming that the equivalence of steady-state endocytic trafficking dynamics between untagged EGFR and EGFR-GFP.

To test whether the EGFR-GFP localization observed in flank tumor xenografts is common with a physiologically relevant model for head and neck carcinoma, we also performed equivalent experiments using tongue orthotopic xenograft (*Szaniszlo et al., 2014*; *Amornphimoltham et al., 2017*). HSC3/EGFR-GFP cells were implanted in the tongue of athymic nude mice, and after 3 weeks, tumors were dissected, fixed, sectioned and analyzed by confocal microscopy. Montage images covering large areas of tumors revealed predominant localization of EGFR-GFP at the cell surface with occasional clusters of GFP in vesicles, a distribution pattern essentially similar to that observed in flank tumors (*Figure 2D*). Together, the data in *Figures 2* and *3A* strongly suggest that only a small fraction of EGFRs are endocytosed under steady-state growth conditions of HSC3/EGFR-GFP tumor xenografts (in the presence of endogenous EGFR ligands).

## Small pool of active EGFRs is predictive of picomolar ligand concentrations in tumors

The finding of a relatively low incidence of detectable EGFR-GFP endocytosis in HSC3 tumor xenografts (*Figures 2* and *3A*) suggests that either the concentration of ligands accessible to EGFR in these tumors is quite low, resulting in activation of only a small pool of EGFRs, or EGFRs are incapable of efficient endocytosis in vivo. To test the first hypothesis, we measured EGFR activity in tumors by determining the levels of EGFR tyrosine phosphorylation and compared the levels of EGFR phosphorylation in tumors with those measured in vitro in the same HSC3/EGFR-GFP cells treated with a range of EGF concentrations. The reasons for focusing on EGFR activity rather than carrying out direct measurements of ligand concentrations are two-fold. First, receptor activity is the key parameter for defining down-stream signaling events. Second, it is not feasible to measure concentrations of all seven EGFR ligands, and some ligand detection kits do not differentiate between soluble and immature transmembrane ligands.

In vitro, phosphorylation of Tyr1068 in EGFR and EGFR-GFP (measured as the pY1068/EGFR ratio) displayed a linear dependence on the concentration of EGF when incubated with cells for 10 min (*Figure 4A*). When cells were incubated with the same range of EGF concentrations for 12 hr, the extent of Tyr1068 phosphorylation mirrored the results obtained using an acute stimulation (*Figure 4A*). Furthermore, ligand dose-response curves were similar in cells stimulated acutely (10 min) with EGF and transforming growth factor alpha (TGFα), an EGFR ligand that is frequently produced by cancer cells (*Grandis and Tweardy, 1993*) but is more readily dissociated from the receptor in the acidic environment of endosomes when compared with the canonical ligand, EGF (*Roepstorff et al., 2009*) (data not shown). Therefore, EGF-dose-dependence of the pY1068/EGFR ratio measured using a 10-min stimulation of cells in vitro was employed as a linear standard to estimate ligand concentrations in vivo in tumor xenografts. In the example shown in *Figure 4B–C*, the pY1068/EGFR ratio measured in two tumor xenograft samples (last two lanes in *Figure 4B*) was interpolated into the corresponding in vitro curve to estimate the EGFR ligand concentration in these two tumors. Such an estimation of ligand concentration based on the in vitro/in vivo comparison of EGFR phosphorylation was performed in multiple flank and tongue xenografts of HSC3/EGFR-GFP cells in several independent series of experiments (*Figure 4D–E*). The analysis yielded mean predicted concentrations of EGFR ligands as low as ~0.2 and ~0.6 ng/ml (~34 and 100 pM) in the flank and tongue implants, respectively (*Figure 4D–E*). It should be emphasized that this estimation is based on the assumption that all cells in vitro and in vivo have access to EGFR ligands. This assumption is technically difficult to test directly with high precision, although immunolabeling of

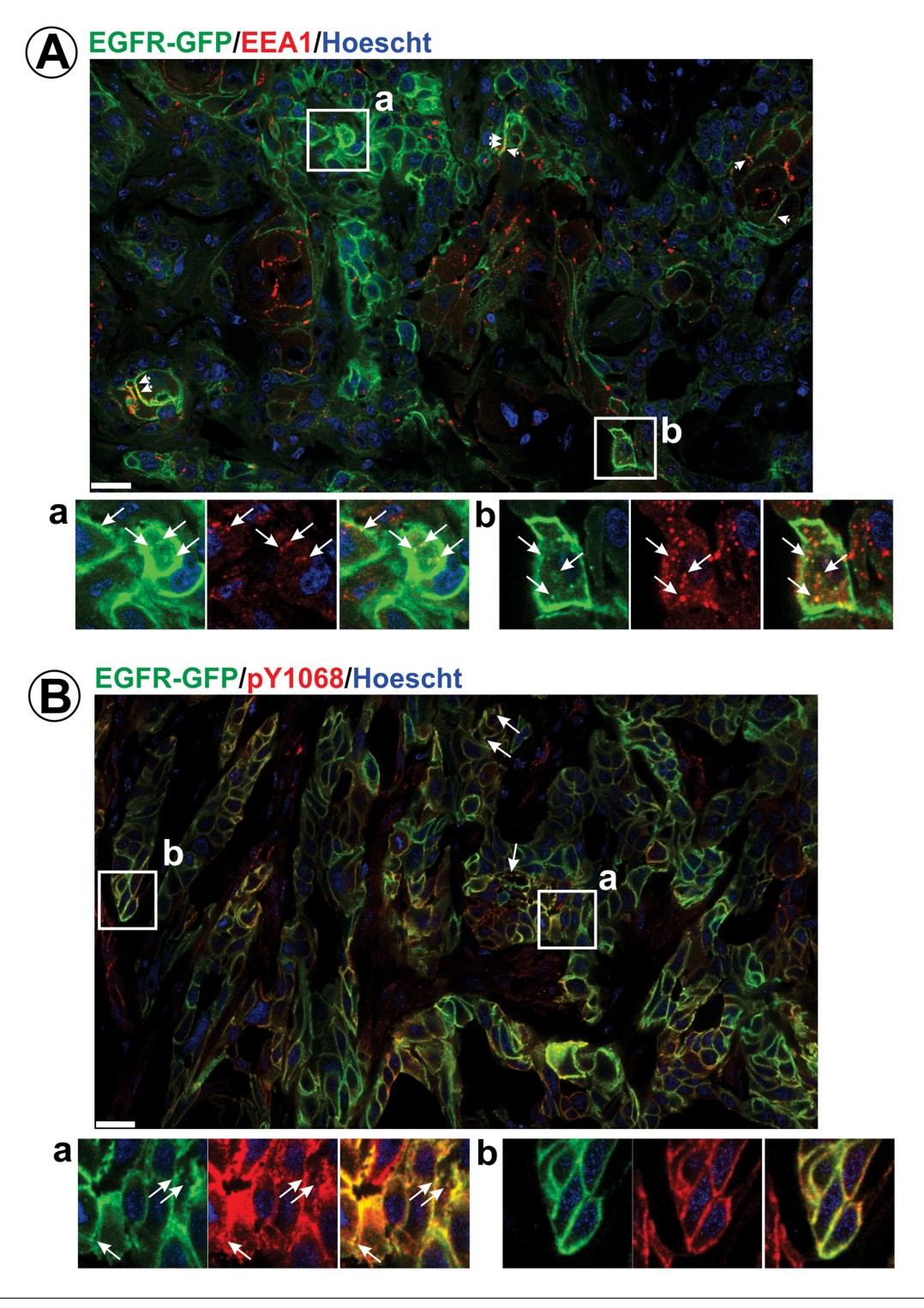

**Figure 3.** Immunofluorescence labeling of early endosomes and phosphorylated EGFR in HSC3/EGFR-GFP flank tumor xenografts. HSC3/EGFR-GFP tumors were dissected and fixed. Cryosections were permeabilized with Triton X-100 and immunolabeled with EEA1 (**A**) or pY1068 antibodies (**B**). Nuclei were stained with Hoescht33342. Confocal images were acquired through 405 nm (Hoescht; *blue*), 488 nm (EGFR-GFP; *green*) and 640 nm (Cy5-conjugated secondary antibody; *red*) channels. Montage images of representative large areas of tumors are shown. Insets show high-magnification images of regions indicated by white rectangles. Long arrows indicate examples of vesicles containing EGFR-GFP co-localized with EEA.1 or pY1068 immunofluorescence. Short arrows

*Figure 3 continued on next page*

*Figure 3 continued*
point on examples of EEA.1 endosomes overlapping with the plasma membrane. Scale bars, 25 µm. Images
demonstrating the specificity of EEA.1 and pY1068 labeling are presented in *Figure 3—figure supplement 1*.
DOI: https://doi.org/10.7554/eLife.31993.007
The following figure supplement is available for figure 3:

**Figure supplement 1.** Controls for specificity of EEA.1 and pY1068 labeling.
DOI: https://doi.org/10.7554/eLife.31993.008

flank tumor xenograft sections with the pY1068 antibody demonstrated EGFR phosphorylation in almost all cells in tumor sections with the predominant localization of the phosphorylated receptor at the plasma membrane (*Figure 3B* and *Figure 3—figure supplement 1B*). These results indicate that EGFRs are certainly able to bind to ligands in a large fraction of cells in vivo. The intensity of pY1068 signal was proportional to the intensity of GFP (EGFR-GFP) signal, suggesting the relatively homogenous activation of EGFR within the tumor.

To determine whether our observations in oral squamous carcinoma implants are common to other types of cancer, NSCLC H322 cells and triple negative breast cancer MDA-MB-468 cells were implanted into the flank of athymic nude mice. Both these cell lines express wild-type EGFR, and their tumor xenografts are shown to be growth-dependent on the EGFR activity (*Smith et al., 2015*; *Busser et al., 2010*; *El Guerrab et al., 2016*). The comparative in vitro/in vivo analysis of the EGFR activity, performed as with HSC3 cells, estimated low EGFR ligand concentrations in xenograft tumors of H322 and MDA-MB-468 cells, although these concentrations displayed a wider range in MDA-MB-468 tumors (*Figure 4D–E*). In summary, the in vitro/in vivo calibration analysis of the EGFR activity predicted that mean concentrations of EGFR ligands in tumor xenografts are 0.1–0.6 ng/ml (17–100 pM). Importantly, pY1068/EGFR ratio values measured in human HNSCC specimens were within the range of those values found in mouse tumor xenografts of HSC3/EGFR-GFP cells (*Figure 4F–G*), suggesting the presence of low EGFR ligand concentrations in human HNSCC.

## ERK1/2 pathway is the primary signaling axis that is significantly activated by picomolar EGFR ligand concentrations

As our analysis predicts low EGFR ligand concentrations in tumor xenografts in vivo, we examined which signaling cascades are activated by the same EGF concentrations in cultured cells in vitro. To this end, EGF-dose-response to activation-dependent phosphorylation of several signaling proteins was measured using western blots. While EGFR phosphorylation at Tyr1068 displayed linear dependence on EGF concentration in the range of 0–5 ng/ml EGF (*Figure 5A*), MEK1/2 and ERK1/2 phosphorylation reached 75–80% of the maximum phosphorylation in cells stimulated with 0.5 ng/ml (85 pM) EGF (*Figure 5B*). In contrast, tyrosine phosphorylation of phospholipase C γ1 was marginally stimulated by 0.5–1 ng/ml EGF and increased proportionally to the EGF concentration, in a manner comparable to that of the pY1068 signal (*Figure 5C*). Likewise, significant phosphorylation of STAT3 was reliably detected only at >1 ng/ml EGF and was also proportional to EGF concentrations (*Figure 5D*). Constitutive activity of AKT (*Figure 5E*) and SFKs measured by phosphorylation of their catalytic tyrosines (*Figure 5F*) was high, and EGF stimulation did not lead to a substantial increase of these activities. Similar patterns of EGF-dose-dependence of the major EGFR signaling pathways have been previously observed in several other types of cultured cells (*Shi et al., 2016*; *Krall et al., 2011*).

These in vitro data prompted us to hypothesize that the regulation of EGFR-dependent tumor growth in vivo is predominantly driven by signaling through the ERK1/2 pathway. ERK1/2 phosphorylation in tumors normalized to this phosphorylation in vitro (at 1 ng/ml EGF) was comparable with the normalized phosphorylation of EGFR at Tyr1068 (*Figure 5G*). By contrast, in vivo/in vitro ratios of phosphorylation of other signaling effectors, especially STAT3, were significantly higher than these ratios of EGFR and ERK1/2 (*Figure 5G*), which may be attributable to EGFR-independent activation of the former pathways in HSC3/EGFR-GFP cells in tumors in vivo, and possibly, detection of phosphoproteins in mouse cells contaminating tumor samples. Overall, the data in *Figure 5* support the model whereby the ERK1/2 pathway is primarily EGFR-driven in vivo and important for EGFR-

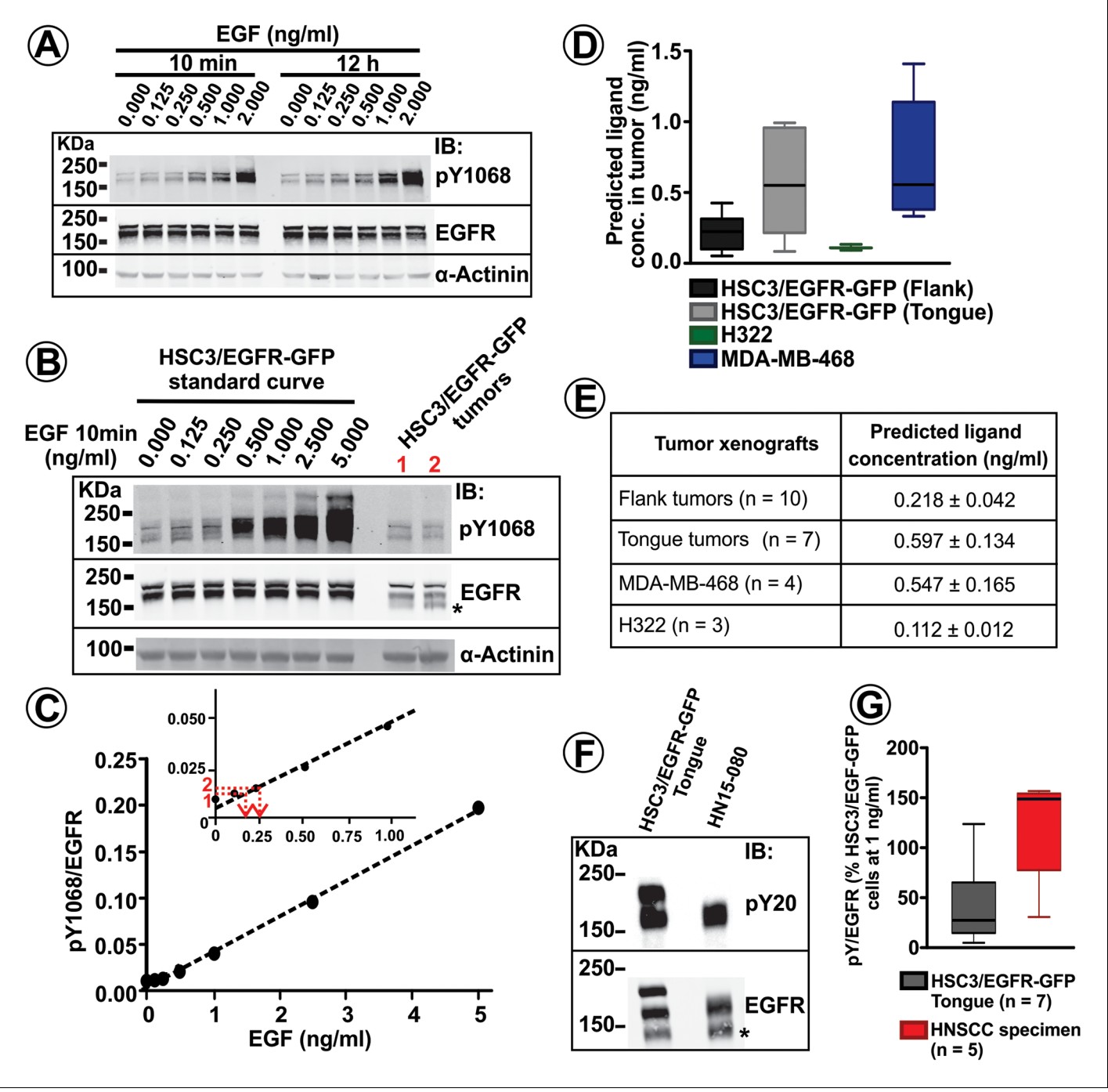

**Figure 4.** Quantification of EGFR phosphorylation and EGFR ligand concentrations in tumor xenografts and human HNSCC specimens. (**A**) Serum-starved HSC3/EGFR-GFP cells were treated with 0–2 ng/ml EGF for 10 min or 12 hr at 37°C as described in 'Materials and methods'. Lysates were probed with pY1068, EGFR and α-actinin (loading control) antibodies by Western blotting. Representative blot is shown. (**B**) Lysates of HSC3/EGFR-GFP cells treated with EGF (0–5 ng/ml) for 10 min at 37°C and two HSC3/EGFR-GFP flank tumors (#1 and #2; obtained as in *Figure 1F*) were probed with pY1068, EGFR and α-actinin (loading control) antibodies by western blotting. Representative blot is shown. (**C**) The values of in vitro pY1068/EGFR ratio measured in the experiment presented in (**B**) are plotted against EGF concentration. pY1068/EGFR ratios from tumors #1 and #2 were interpolated in the plot to estimate EGFR ligand concentrations in these two tumors (see red lines in the inset). (**D and E**) Summary of the quantifications of predicted EGFR ligand concentrations in HSC3/EGFR-GFP, H322 and MDA-MB-468 tumor xenografts from several independent series of experiments performed as described in (**B–C**). Boxplot in (**D**) shows medians, quartiles, and extreme values (n = 3–10). Mean values (±S.E.M.) are presented in (**E**). (**F**) EGFR was immunoprecipitated from the HSC3/EGFR-GFP tongue tumor and the HNSCC patient specimen. Immunoprecipitates were blotted with pY20 and

*Figure 4 continued on next page*

Figure 4 continued

EGFR antibodies. Asterisk indicates EGFR calpain proteolytic product (~145 kDa). (**G**) Quantification of pY20/EGFR ratios in tongue tumors and HNSCC patient specimens from several experiments exemplified in (**F**). Boxplot shows medians, quartiles, and extreme values (n = 4–5).

DOI: https://doi.org/10.7554/eLife.31993.009

dependent tumorigenesis, whereas multiple other signaling pathways are largely involved in EGFR-independent tumor growth.

## EGFR-GFP endocytosis visualized by fluorescent EGF in HSC3/EGFR-GFP tumor xenografts

The experiments described in *Figures 1–4* provide ample evidence to support the hypothesis that low concentrations of endogenous EGFR ligands accessible to cells in tumor xenografts explain our observation of a small pool of detectable intracellular EGFR-GFP vesicles. To test the alternative hypothesis that EGFRs are essentially incapable of an efficient endocytosis in tumors in vivo, EGF-Rh was injected intravenously into mice harboring HSC3/EGFR-GFP tumor xenografts. EGF-Rh could be detected in flank and tongue tumor xenografts ~1 hr after tail vein injection. The pY1068/EGFR ratio in flank tumors was increased 2.5-fold after EGF-Rh administration (*Figure 6A*). Considering the range of endogenous ligand concentrations predicted in *Figure 4*, the total concentration of EGFR ligands in the presence of systemic EGF-Rh is estimated to be lower than 1–2 ng/ml.

Intravital imaging of flank tumor xenografts after EGF-Rh injection revealed co-localization of EGFR-GFP and EGF-Rh in perinuclear vesicles and clusters located proximally or overlapping with the plasma membrane (*Figure 6B*). Similar to that observed in the absence of EGF-Rh, time-lapse imaging showed that the motility of EGF-Rh/EGFR-GFP vesicles is very limited when compared with what is typically observed in vitro. Imaging of EGF-Rh and EGFR-GFP fluorescence on flank tumor sections demonstrated the presence of vesicles containing both ligand and receptor in a large population of cells (*Figure 6C*). Quantitation of cells containing EGF-Rh-positive vesicles on multiple sections from different tumors revealed that EGF-Rh had access to EGFR in at least 40–50% of the cells in these tumors. In most cells, EGF-Rh was detected predominantly in endosomes and was essentially absent in the plasma membrane, indicative of a fast and efficient endocytosis (*Figure 6C*). The accumulation of EGF-Rh in endosomes was increased in clusters of cells at the periphery of tumor nodules, presumably in the vicinity of blood vessels (see large area montage image in *Figure 6—figure supplement 1*). Co-staining of these sections with EEA.1 antibody demonstrated that many of these vesicles are early/sorting endosomes (*Figure 6—figure supplement 2A*). Endosomes containing EGF-Rh and EGFR-GFP were also co-labeled with the pY1068 antibody, confirming receptor activity in endosomes (*Figure 6—figure supplement 2B*).

A similar localization pattern for EGF-Rh:EGFR-GFP complexes was observed in tongue tumors, although a lower number of EGF-Rh-labeled cells was detected, probably, due to a strong autofluorescence of muscle cells in the red channel interfering with the detection of weak endosomal EGF-Rh signals (*Figure 6D*). Comparison of EGF-Rh and EGFR-GFP localization in vivo with that in HSC3/EGFR-GFP cells stimulated with 0.5–10 ng/ml EGF-Rh in vitro demonstrated the similarity of the in vivo pattern with that in cells treated in vitro with 0.5–1 ng/ml EGF-Rh (predominant accumulation in endosomes but not at the cell surface; see *Figure 6E*). Collectively, these data (*Figure 6* and *Figure 6—figure supplements 1–2*) show that EGFR is capable of rapid internalization in the majority of cells in tumor xenografts in the presence of low concentrations of endogenous and exogenous EGFR ligands.

## EGFR endocytosis is kinase-dependent and uses the CME pathway in tumors in vivo

The demonstration of an efficient EGF-Rh endocytosis in vivo allows us to start to unravel the underlying mechanism(s) of this endocytosis. First, we tested whether EGFR kinase activity is necessary for receptor internalization in vivo by acutely treating mice harboring HSC3/EGFR-GFP flank tumors with gefitinib. 2 hr after i.p. injection of gefitinib, EGF-Rh was injected i.v., and the localization of EGF-Rh and EGFR-GFP was analyzed on tumor sections. Gefitinib treatment resulted in a strong inhibition of EGF-Rh endocytosis as evidenced by the dramatic reduction in the proportion of cells with

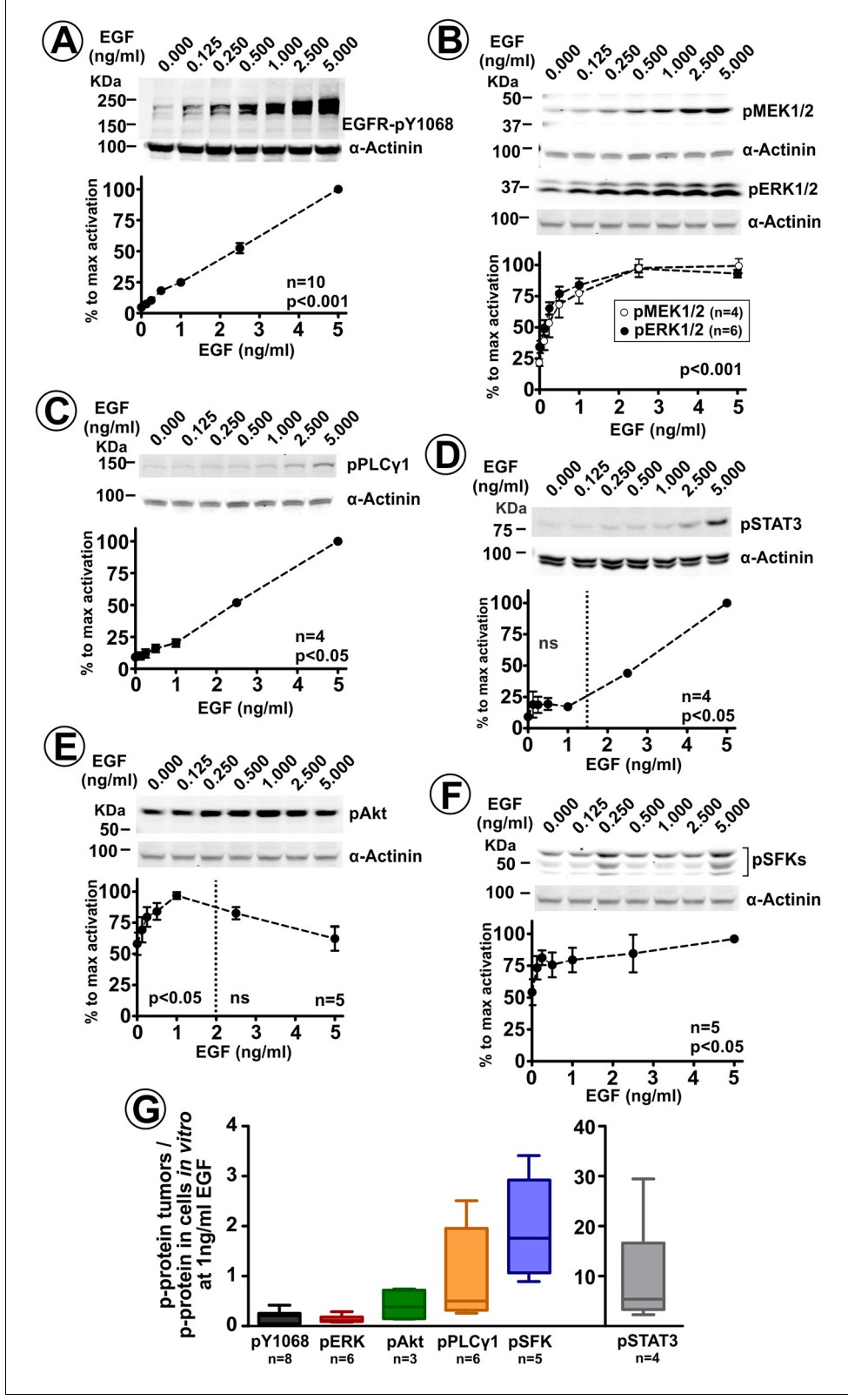

**Figure 5.** ERK1/2 pathway is the primary signaling cascade significantly activated by low ligand concentrations. Serum-starved HSC3/EGFR-GFP cells were incubated with 0–5 ng/ml EGF for 10 min at 37°C. Lysates were probed with antibodies to phosphorylated proteins: EGFR pY1068 (A), MEK1/2 and ERK1/2 (B), PLCγ1 (C), STAT3 (D) AKT (E) and SFKs (F). Blotting for α-actinin is used as a loading control in each experiment. Representative western

*Figure 5 continued on next page*

*Figure 5 continued*

blots and quantifications of several independent experiments are shown. Paired T-test was performed (n = 4–10). p values indicate statistical significance relative to 'no EGF' cells. *ns*, p>0.05. (G) Lysates of HSC3/EGFR-GFP flank and tongue tumors, and lysates of these cells grown in vitro and treated with 0–5 ng/ml EGF for 10 min at 37°C (as in the experiment exemplified in *Figure 4B*) were blotted with phosphosite-specific antibodies as in (A–F). Phosphosite antibody signals (*p-Protein*) were normalized to the amounts of α-actinin (loading control). Normalized signals of each phosphosite antibody in tumor lysates ('in vivo') were divided by normalized signal intensities of the same antibody in lysates of cultured cells treated with 1 ng/ml EGF in vitro. Boxplot shows medians, quartiles, and extreme values of the resulting 'in vivo/in vitro' ratios of normalized signals (n = 2–8).
DOI: https://doi.org/10.7554/eLife.31993.010

detectable EGF-Rh/EGFR-GFP-containing endosomes when compared to vehicle-treated tumors (*Figure 7*). This result demonstrated that in vivo EGFR endocytosis is receptor-kinase-dependent.

Strong accumulation of EGF-Rh in endosomes and kinase-dependency of EGFR endocytosis observed in vivo are characteristic of the receptor internalization through CME as observed in vitro under conditions when low ligand concentrations are used (*Chen et al., 1989*). Therefore, a cocktail of small molecule inhibitors of CME, Dyngo-4a and Pitstop2, was administered i.p. in mice harboring flank tumors, followed by the EGF-Rh internalization assay 2 hr later. These inhibitors substantially decreased the number of cells with detectable EGF-Rh/EGFR-GFP-containing endosomes (*Figure 7*). This result indicates that CME is a primary internalization route of ligand-occupied EGFR in tumors in vivo.

## EGFR is ubiquitylated in tumors in vivo but is unaffected by Cbl overexpression

Detection of EGFR phosphorylation at Tyr1068 (*Figure 1*), a major Grb2-binding site necessary for Cbl-mediated ubiquitylation of the receptor (*Waterman et al., 2002*; *Jiang et al., 2003*) suggested that EGFR could be ubiquitylated in vivo, and that Cbl proteins may, therefore, regulate EGFR traffic and activity in tumors. To examine EGFR ubiquitylation, EGFR and EGFR-GFP were immunoprecipitated from HSC3/EGFR-GFP tumor lysates, and the extent of ubiquitylation of immunoprecipitated EGFRs was compared with that in the same cells in vitro using western blotting. These experiments demonstrated EGFR ubiquitylation in both flank and tongue HSC3/EGFR-GFP tumors (*Figure 8A–B*). The extent of EGFR ubiquitylation in vivo roughly corresponded to that in cells stimulated with 0.5–1 ng/ml EGF in vitro (*Figure 8A–B*). Moreover, comparable levels of EGFR ubiquitylation were detected in specimens of human HNSCC expressing EGFR (*Figure 8B*).

To test whether increased EGFR ubiquitylation results in attenuation of the EGFR signaling and tumorigenesis, HSC3/EGFR-GFP cells overexpressing c-Cbl under tet-inducible promoter were generated. In these cells, c-Cbl concentration was increased more than 100-fold by doxycycline (*Figure 8—figure supplement 1A*). c-Cbl overexpression in cultured cells led to down-regulation of the EGFR protein and increased ubiquitylation of the residual receptor when high (10–100 ng/ml) concentrations of EGF were used (*Figure 8—figure supplement 1B*). By contrast, no significant difference in EGFR levels was detected between control and c-Cbl overexpressing cells incubated with 1 ng/ml EGF (*Figure 8—figure supplement 1B*).

Increased c-Cbl levels were also dramatically increased by doxycycline in flank tumor xenografts of HSC3/EGFR-GFP/tet-Cbl cells in vivo (*Figure 8D*). However, surprisingly, EGFR ubiquitylation and phosphorylation, as well as tumor growth rates were not significantly affected by Cbl overexpression (*Figure 8C–D*). The data in *Figure 8* and *Figure 8—figure supplement 1* further demonstrate the similarity of EGFR behavior in tumors and in cultured cells exposed to low EGF concentrations.

## Discussion

The present study represents the first high-resolution microscopic analysis of endogenous EGFR endocytosis combined with the first quantitative analysis of EGFR phosphorylation and ubiquitylation in EGFR-dependent tumors in vivo. The key technical advance that enabled these analyses was the generation of human cancer cells expressing endogenous GFP-tagged receptor, which eliminates potential confounding effects of EGFR overexpression commonly seen when using constitutive

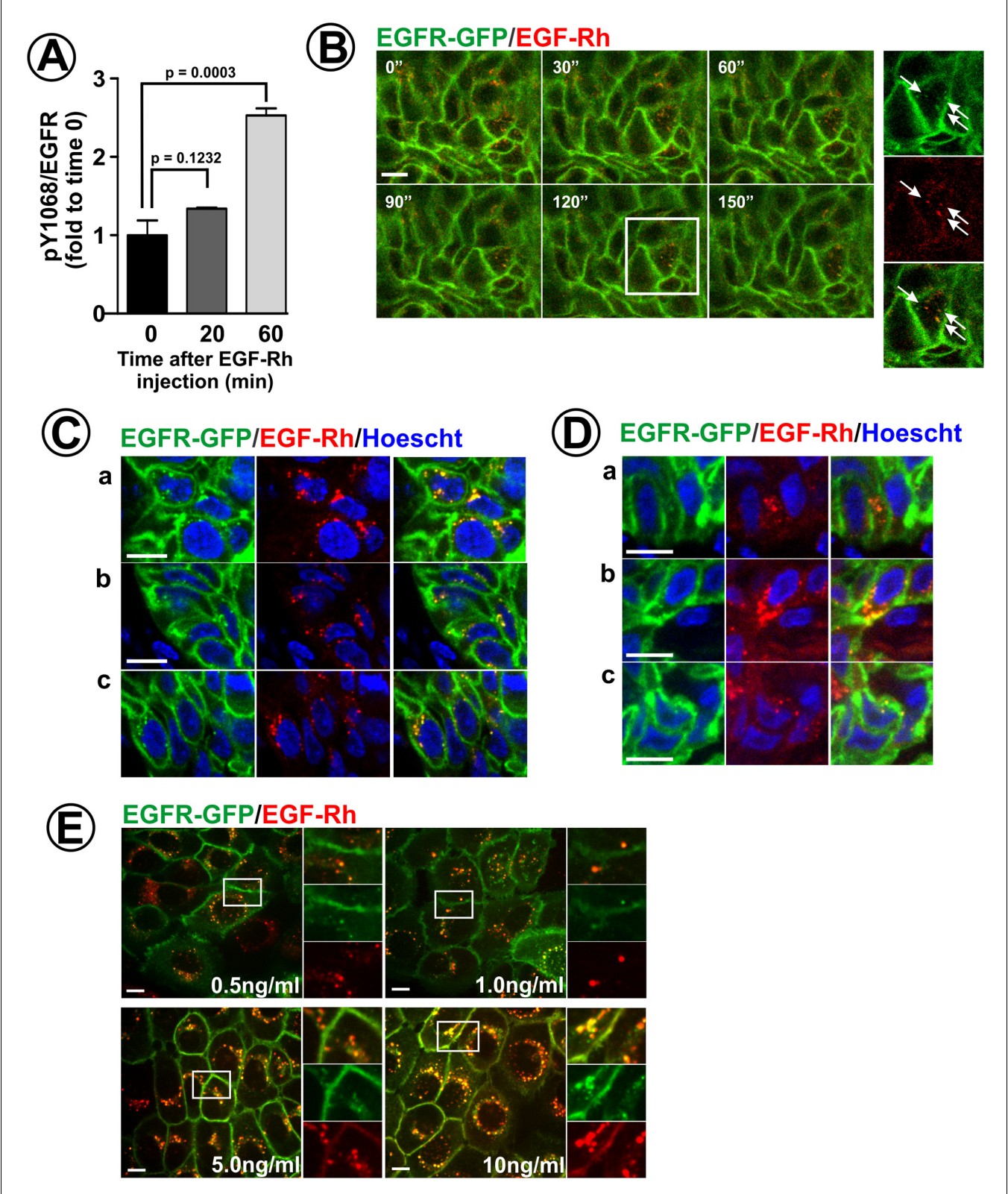

**Figure 6.** Endocytosis of EGF-Rh in flank and tongue tumors. (**A**) Athymic nude mice harboring HSC3/EGFR-GFP xenografts were injected with EGF-Rh (100 µl, 50 ng/µl) in the tail vein. Tumors were dissected before and 20–60 min after injection, and tumor lysates were probed with pY1068 and EGFR antibodies as in *Figure 4A–B*. The mean values of pY1068/EGFR ratios (±S.E.M) are presented as fold increase to the ratio values before injection of EGF-Rh (time '0'). Unpaired T-test was performed (n = 4). p values < 0.05 are considered statistically significant. (**B**) Mice harboring HSC3/EGFR-GFP

*Figure 6 continued on next page*

*Figure 6 continued*

xenografts were injected with EGF-Rh as in (**A**). Time-lapse intravital imaging of GFP and rhodamine was performed as in *Figure 2A*. Merged images of selected time points are presented. *Insets* show high-magnification single-channel images of the region indicated by white rectangle to demonstrate an overlap of EGF-Rh and EGFR-GFP fluorescence in endosomes (arrows). Time '0' corresponds to the start of time-lapse imaging. Scale bar, 10 μm. (**C** and **D**) Mice harboring flank (**C**) or tongue (**D**) HSC3/EGFR-GFP xenografts were injected with EGF-Rh as in (**A**), tumors were dissected 1 hr after injection and fixed in paraformaldehyde. Confocal imaging of cryosections was performed through 405 nm (Hoescht; *blue*), 488 (EGFR-GFP; *green*) and 561 nm (EGF-Rh; *red*) channels. Representative images are shown. Scale bars, 10 μm. Regions corresponding to these images are marked by white rectangles in the montage images of the large areas of tumors presented in *Figure 6—figure supplement 1*. (**E**) HSC3/EGFR-GFP cells grown in vitro were stimulated with EGF-Rh (0.5–10 ng/ml) for 15 min at 37°C, Confocal imaging was performed through 488 nm (EGFR-GFP; *green*) and 561 nm (EGF-Rh; *red*) channels. The image acquisition parameters were the same as in (**C and D**). *Insets* show high-magnification single-channel images of the regions indicated by white rectangles to demonstrate EGF-Rh remaining at the cell surface when used at 5–10 ng/ml but not at 0.5–1 ng/ml. Scale bars, 10 μm.

DOI: https://doi.org/10.7554/eLife.31993.011

The following figure supplements are available for figure 6:

**Figure supplement 1.** Endocytosis of EGF-Rh in flank and tongue tumors.

DOI: https://doi.org/10.7554/eLife.31993.012

**Figure supplement 2.** Immunofluorescence labeling of early endosomes and phosphorylated EGFR in HSC3/EGFR-GFP tumor flank xenografts in the presence of EGF-Rh.

DOI: https://doi.org/10.7554/eLife.31993.013

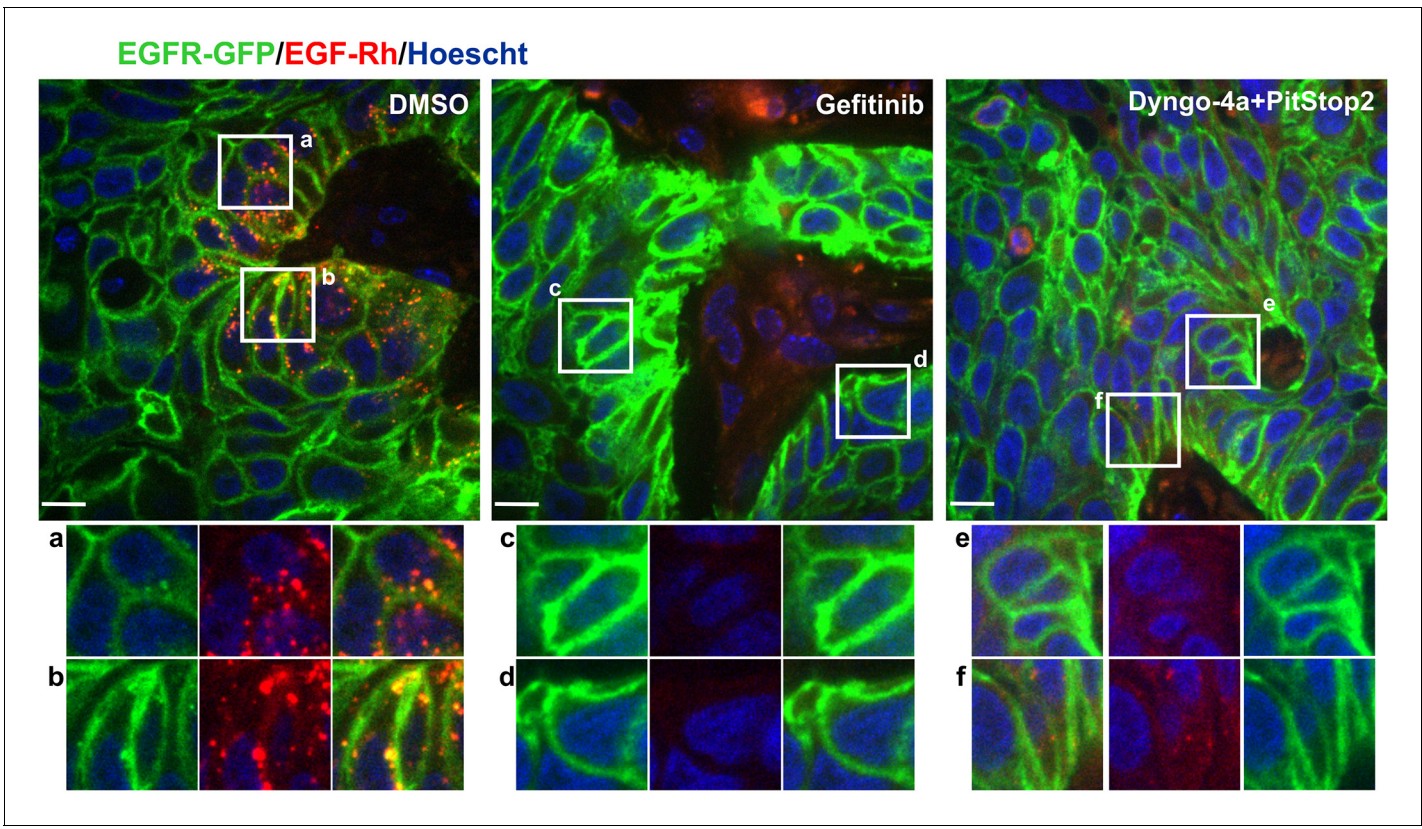

**Figure 7.** Effects of gefitinib and endocytosis inhibitors on EGF-Rh and EGFR-GFP internalization in tumors. Mice harboring flank HSC3/EGFR-GFP xenografts were administered i.p. with DMSO, gefitinib (30 mg/kg) or Dyngo-4a plus Pitstop2 (1.125 mM/each in 400 μl saline-glucose). 2 hr after these injections, EGF-Rh was i.v. injected as in *Figure 6*. Tumors were dissected 1 hr after EGF-Rh injection and fixed in paraformaldehyde. Confocal imaging of cryosections was performed through 405 nm (Hoescht; *blue*), 488 nm (EGFR-GFP; *green*) and 561 nm (EGF-Rh; *red*) channels. Scale bars, 10 μm. Representative merged images are shown. *Insets* show high-magnification single-channel images of the regions indicated by white rectangles. Intensity scales are identical.

DOI: https://doi.org/10.7554/eLife.31993.014

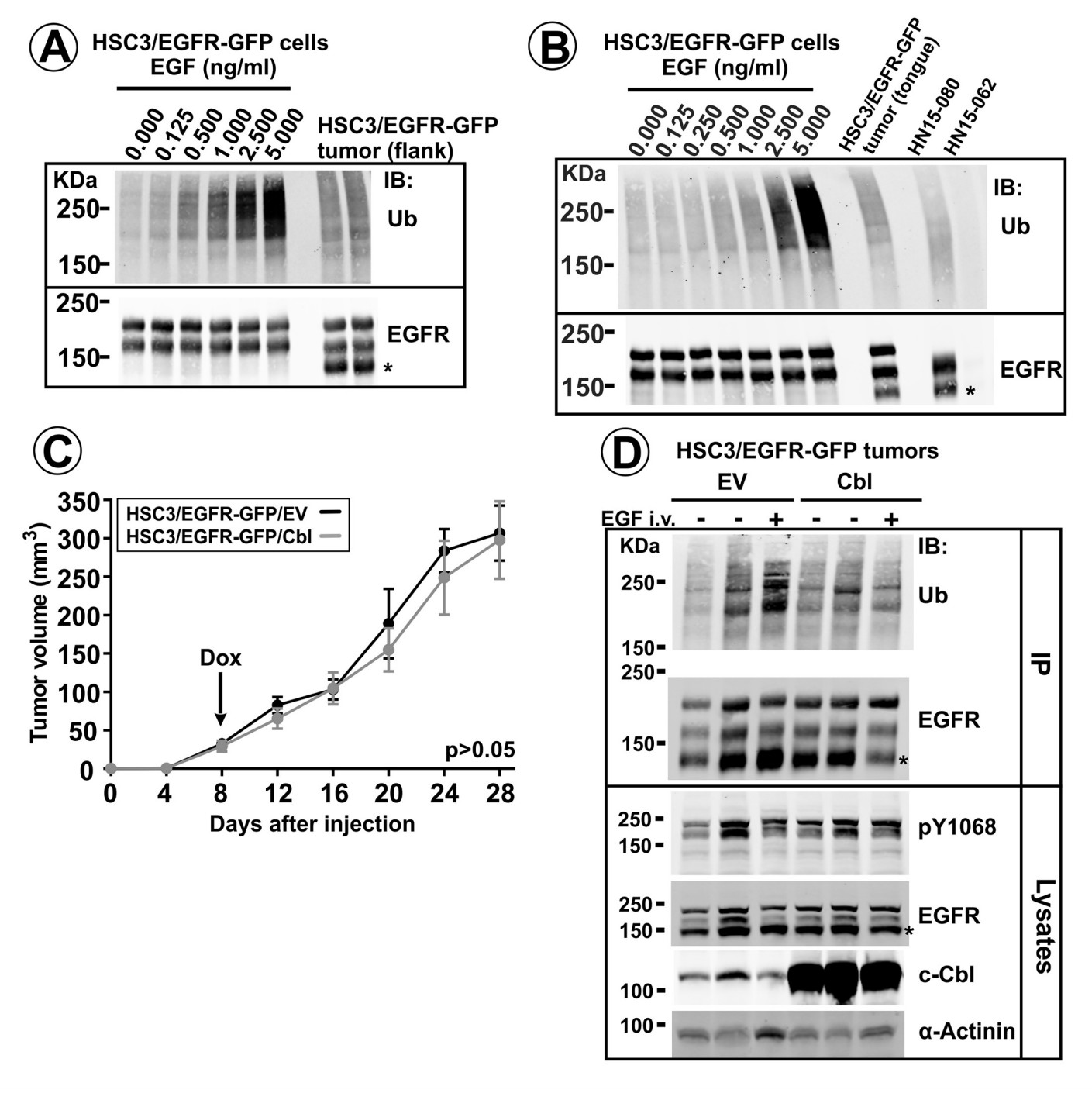

**Figure 8.** EGFR ubiquitylation in tumors in vivo, and the effects of Cbl overexpression. EGFR/EGFR-GFP were immunoprecipitated from lysates of HSC3/EGFR-GFP cells grown in vitro and stimulated with 0–5 ng/ml EGF for 10 min (**A–B**), lysates of flank (**A**) and tongue (**B**) HSC3/EGFR-GFP tumors or lysates of human HNSCC specimens (**B**). Immunoprecipitates were analyzed by western blotting with ubiquitin and EGFR antibodies. Representative western blots are shown. Asterisk indicates EGFR calpain proteolytic product (~145 kDa). Tumor specimen HN15-062 expressing negligible amount of EGFR is shown as control for the specificity of the ubiquitylated EGFR signal. (**C and D**). Mice harboring HSC3/EGFR-GFP/Cbl and HSC3/EGFR-GFP/EV (control) flank tumors for 8 days were switched to drinking water containing 0.5 mg/ml Dox to induce c-Cbl overexpression. (**C**) Tumor growth was monitored as described in 'Methods'. Averaged tumor volumes (±S.E.M; n = 9) are presented. Paired T-test was performed. (**D**) After 28 days of Dox administration, three tumors (one – 1 hr after EGF-Rh injection) were dissected and lysed. Aliquots of the lysates were probed with pY1068, EGFR, c-Cbl and α-actinin antibodies (loading control), or the rest of lysates was used for immunoprecipitation of EGFR and EGFR-GFP. Immunoprecipitates were probed with ubiquitin and EGFR antibodies.

*Figure 8 continued on next page*

*Figure 8 continued*

DOI: https://doi.org/10.7554/eLife.31993.015

The following figure supplement is available for figure 8:

**Figure supplement 1.** Effect of c-Cbl overexpression on EGFR levels and ubiquitylation in vitro.

DOI: https://doi.org/10.7554/eLife.31993.016

promoters, and permits the use of intravital imaging of receptor dynamics in living tumors and direct fluorescence imaging of fixed tumors. During preparation of our manuscript, Yang and co-workers (*Yang et al., 2017*) reported generation of a knock-in mouse expressing EGFR tagged with the Emerald fluorescent protein, which was exploited to study the localization of EGFR in normal mouse tissues and carcinogen-induced colon adenocarcinoma. The latter study and our experiments illustrate the advantage of the direct fluorescence imaging of EGFR as compared with imaging using immunofluorescence approaches with its inherently lower signal-to-noise ratio and the potential to not label hidden or complexed epitopes. Additionally, having membranes of tumor cells labeled with EGFR-GFP in our experimental system allowed clear separation of tumor boundaries from the surrounding mouse tissue and stroma.

Overall, combining intravital and confocal imaging with the comparative in vivo:in vitro biochemical analyses allowed us to conclude that the EGFR physiology in EGFR-dependent tumor xenografts displays all characteristics observed in cells grown in vitro and exposed to low (<1–2 ng/ml) EGFR ligand concentrations. The prediction of low EGFR ligand concentrations in tumors in vivo is based on comparing: (1) apparent levels of EGFR phosphorylation (*Figure 4*) and ubiquitylation (*Figure 8*) in vivo and in vitro; (2) patterns of EGFR-GFP and EGF-Rh localization, and mechanism(s) of their endocytosis in vivo and in vitro (*Figure 6*); and (3) effects of c-Cbl overexpression on EGFR levels, ubiquitylation in vitro and in vivo, and tumor growth (*Figure 8*).

Our principal approach was based on measuring EGFR phosphorylation (as a direct measure of its activation) and other receptor activities downstream of, and dependent on, ligand binding, instead of attempting to measure intra-tumoral concentrations of multiple EGFR ligands. Such measurements are particularly challenging because autocrine produced ligands are rapidly depleted from the extracellular fluids by binding to EGFRs and endocytosis (*DeWitt et al., 2001*). Tumor cells have access to circulating ligands that induce paracrine activation of EGFR, and in addition, some tumor cells produce augmented amounts of EGFR ligands, which initiate signaling in an autocrine fashion (*Hobor et al., 2014*; *Liu et al., 2016*; *Zhou et al., 2014*). Tumorigenesis may elevate ligand concentrations by altering the expression or surface maturation of EGFR ligands (*Yamane et al., 2008*). For example, increased production of TGFα, amphiregulin or heparin-binding EGF-like growth factor was observed in some tumors (*Grandis and Tweardy, 1992*; *Grandis and Tweardy, 1993*; *Yagi et al., 2005*; *Li et al., 2010*; *Normanno et al., 2001*; *Vermi et al., 2013*). EGF concentration in tumors may also be increased by the paracrine mechanism, for example, through EGF production by tumor-associated macrophages (*Goswami et al., 2005*). Therefore, variations from the ligand concentration range predicted in our studies are possible in tumors simply due to varying stages of development and vascularization. However, as we found that predicted EGFR ligand concentrations were essentially similar in HCS3/EGFR-GFP flank xenografts harvested 1–5 weeks old after implantation and within the range of tumor volumes of 100–700 mm$^3$ (data not shown), such deviations are unlikely to significantly elevate ligand concentrations. We did find that ligand concentrations in tongue tumors can be slightly higher than in flank xenografts (*Figure 4*), probably, because of the tumor proximity to submandibular glands, the main site of mouse EGF synthesis.

It should be emphasized that our measurements give a range of ligand concentrations in tumors (mean values: 17–100 pM) and do not provide the precise concentrations. These estimations are based on the assumption that the same fraction of cells is accessible to ligands in vitro and in vivo. Based on quantification of the experiments with EGF-Rh (*Figure 6*), at least ~50% of cells in the tumor are accessible to the circulating ligands. Assuming that in vitro all cells have access to the ligand, the actual ligand concentrations in vivo may be two-fold higher than those predicted in *Figure 4*. However, the fraction of cells containing EGF-Rh-bound receptors in thin cryostat sections may be an underestimate because cells, that contain labeled endosomes situated out of the section plane, could not be accounted for. Further, the pattern of pY1068 localization suggested that EGFR

is activated in the majority of cells in tumors. Significant variations in the intensity of EGF-Rh (*Figure 6*) and pY1068 fluorescence (*Figure 3* and *Figure 3—figure supplement 1B*) relative to the EGFR-GFP signal have not been observed within the tumor, providing no indications that there is a significant population of cells which are exposed to ligand concentrations that would result in qualitatively different EGFR activation and behaviors. Thus, based on all considerations above, EGFR ligand concentrations in tumor xenografts are predicted to be less than 1–2 ng/ml.

Experiments demonstrating that dramatic overexpression of c-Cbl in vivo did not affect EGFR phosphorylation, ubiquitylation and tumor growth (*Figure 8*), equivalent to cells exposed to low EGF concentrations in vitro, further support the prediction of low concentrations of EGFR ligands and a small pool of active EGFRs in tumors. EGFR ubiquitylation shows a two-phase dependence on EGF concentrations with the threshold at ~1–2 ng/ml EGF (*Figure 8A–B*). At low EGF concentrations, the extent of receptor ubiquitylation may not be sufficient for an efficient targeting of EGFR to lysosomes both in vivo and in vitro as in agreement with that proposed by Sigismund and co-workers (*Sigismund et al., 2008*). Alternatively, the steady-state level of EGFR is maintained in the presence of overexpressed Cbl because degradation of the small pool of ubiquitylated receptors is compensated by de novo receptor biosynthesis.

In summary, EGFR ligand concentrations in tumors predicted in the present study are within the range of $K_D$ values measured for high-affinity EGF binding to EGFR (20–200 pM) (*Ringerike et al., 1998*; *Rees et al., 1984*; *Sorkin et al., 1991*). Therefore, it can be hypothesized that a small pool of high-affinity EGFRs are ligand-occupied in tumor xenografts and sufficient to drive EGFR-dependent tumorigenesis. The importance of the 'high-affinity EGFR' hypothesis is three-fold:

First, EGFR-dependent tumor growth of HSC3 xenografts is driven mainly by the activity of the ERK1/2 signaling pathway because when few EGFRs are activated only the ERK1/2 axis is significantly activated (*Figure 5*). Consistent with the hypothesis of the essential role of ERK1/2 signaling, inhibition of this pathway suppressed the EGFR-dependent growth of tumor xenografts of MDA-MB-468 and other cells (*Chen et al., 2016*; *Zhao and Adjei, 2014*).

Second, a small number of ligand-occupied receptors per cell in tumors predicts that the internalization of activated EGFR is via the CME pathway. In vitro, internalization of 1–4 ng/ml [125]I-EGF was strongly inhibited by siRNA knockdown of clathrin heavy chain in HSC3 cells (data not shown). An efficient internalization of EGF-Rh in tumors in vivo resembles the observations of similarly efficient EGF-Rh CME in vitro (*Figure 6*). The kinase-dependence of EGF-Rh endocytosis in vivo (*Figure 7*) further strengthens the notion of the CME as the primary mechanism for EGFR internalization in tumors in vivo. On the other hand, the results with clathrin/dynamin inhibitors (*Figure 7*) should be considered with caution. Although these compounds and their combination were previously used in in vivo mouse studies (*Joffre et al., 2011*; *Jensen et al., 2017*), it is unclear how much of the compounds actually reach tumor cells, and what the concentrations within the tumor are. The same compounds inhibit EGFR endocytosis in the presence of low EGF concentrations (conditions favoring CME) in vitro (*Pinilla-Macua and Sorkin, 2015*; *Garay et al., 2015*); however, high concentrations of Pitstop-2 and Dyngo-4A are known to inhibit clathrin-independent endocytosis and/or cause general cell toxicity (*Dutta et al., 2012*; *Park et al., 2013*).

Third, the high-affinity EGFR model predicts that even minimal or residual activity of EGFR in human tumors treated with EGFR kinase inhibitors may be sufficient for tumor growth and may therefore underlie tumor resistance to such treatments. Importantly, as shown in our analysis (*Figures 4* and *8*), EGFR phosphorylation and ubiquitylation detected in human HNSCC specimens are within the range of those parameters detected in mouse xenografts, suggesting that the high-affinity model can be applied to human tumors.

Finally, because qualitatively different endocytosis mechanisms and signaling outcomes are observed in cell culture in response to low (<1–2 ng/ml) versus high EGFR ligand concentrations (*Jiang and Sorkin, 2003*; *Sigismund et al., 2005*; *Sigismund et al., 2008*; *Krall et al., 2011*; *Caldieri et al., 2017*), the conclusions and physiological relevance of the vast literature of in vitro studies of EGFR, majority of which utilized high EGF concentrations (20–100 ng/ml), may have to be re-evaluated.

## Materials and methods

### Reagents

Recombinant human EGF was from BD Biosciences. EGF-Rh was from Molecular Probes (Invitrogen). Mouse monoclonal antibody to EGFR phosphotyrosine 1068 (pY1068), phosphorylated ERK1/2, phosphorylated AKT, phosphorylated STAT3, MEK1/2 and c-Cbl; and rabbit polyclonal antibody to ERK1/2, phosphorylated MEK1/2, phosphorylated PLCγ1, phosphorylated SFKs, AKT and α-actinin were from Cell Signaling Technology (Danvers, MA). Polyclonal rabbit antibody to EGFR (1005) and mouse monoclonal to ubiquitin (P4D1) were from Santa Cruz Biotechnology (Dallas, TX). c-Cbl antibody was from BD Bioscience (610442) (San Jose, CA). EGFR-528 monoclonal antibody was from ATCC (Manassas, VA). EEA1 (ab2900) and Y1068 (ab32430) antibodies used for immunostaining of tumor cryosections were from AbCam (Cambridge, MA). Phosphotyrosine-specific antibody pY20 conjugated with horseradish peroxidase (pY20-HRP) was from BD Bioscience (San Jose, CA). Gefitinib was purchased from LC Laboratories (Woburn, MA), whereas Dyngo4a and Pitstop-2 were from AbCam (Cambridge, MA), Hoechst 33342 staining solution was from Thermofisher Scientific (Pittsburgh, PA)

### Cell culture

HSC3 and its derivatives were maintained in DMEM supplemented with 5% fetal bovine serum (FBS). MDA-MB-468 cells were kindly provided by Dr. Oesterreich (Magee-Women's Research Institute, Pittsburgh) and maintained in DMEM:F12 (1:1), 10% FBS, 2 mM L-Glutamine and Penicillin/Streptomycin. H322 cells were obtained from the University of Colorado Cancer Center and maintained in DMEM/10% FBS. The identity of all these lines has been authenticated by STR profiling. These cells are micoplasma-free.

### Generation of HSC3/EGFR-GFP cells

Enhanced obligate heterodimer zinc-finger nucleases (ZFNs) (877 and 878) designed to target ±50 pb of the codon stop of human *EGFR* (*ERBB1*) were engineered by Sigma Life Science (St. Louis, MO). Donor plasmid was designed to carry eGFP fused to the carboxy-terminus of the *EGFR* gene surrounded by ± ~800 pb of homology arms sequences. ZFNs and donor plasmid were transfected into cells using single-cuvette Nucleofector device (Lonza, Allendale, NJ) using the following protocol. Briefly, ~$10^6$ HSC-3 cells were resuspended in Nucleofector solution V and transfected using Nucleofector program P-020. Recovered cells were sorted for GFP-positive signal using an Influx cell sorter directly as single cells into 96-well plates. Positive HSC3/EGFR-GFP clones were transfected for a second time with ZFNs and donor plasmid, and further flow sorted and selected by limited dilution cloning to identify clonal pools with the amount of EGFR-GFP sufficient for detection by live-cell microscopy. The relative concentrations of EGFR-GFP to unlabeled EGFR were determined by PCR and western blotting.

### Generation of cell lines inducibly expressing c-Cbl

The cells expressing c-Cbl under tet-inducible promoter (HSC3/EGFR-GFP/Cbl) were generated using lentiviral infection. A lentiviral construct carrying cDNA encoding human c-Cbl (pSLIK-neo-Cbl vector) was provided by Dr. S. Sigismund (IFOM, Milan, Italy), and lentiviral packaging plasmids were provided by Dr. A. Kwiatkowski (University of Pittsburgh, PA). The lentivirus stock was prepared as described (*Capuani et al., 2015*). HSC3/EGF-GFP cell populations stably expressing pSLIK-neo-Cbl were obtained by selection on neomycin (400 µg/ml). To induce c-Cbl expression, HSC3/EGFR-GFP/Cbl cells were grown for 16 hr, and further grown with 0.5 µg/ml doxycycline for 36 hr, followed by experimental manipulations. Clonal pools carrying an empty pSLIK-neo vector (HSC3-EGFR-GFP/EV) were obtained in parallel and used as an experimental control.

### Tumor xenografts

All experimental procedures involving the use of laboratory mice were approved by the Institutional Animal Care and Use Committee. Athymic nude mice (4- to 5-week-old females) were purchased from Charles Rivers Laboratories Inc. (Wilmington, MA). HSC3/EGFR-GFP, MDA-MB-468 or H322 cell lines were injected s.q. into the flank area of the mice ($1.5 \times 10^6$/100 µl in PBS). Tumor volumes

were measured twice per week using a caliper and the following formula: V = π/6 x (smaller diameter)$^2$ (larger diameter). Tumors were grown to a maximum of 800 mm$^3$. Mice were then euthanized, and tumors were formalin-fixed or snap-frozen in liquid nitrogen.

To study orthotopic tumors, HSC3/EGFR-GFP cells were injected in the tongue of athymic nude mice (3 × 10$^4$/50 µl in PBS). Mice were euthanized after 2 consecutive days of reduced food intake and weight loss, likely, due to tumor interference, were observed. Tumors were formalin-fixed or snap-frozen in liquid nitrogen.

The effect of gefitinib on tumor growth was assessed in flank HSC3/EGFR-GFP tumor-bearing mice. In each experimental series, when tumors reached 100 mm$^3$ (10–16 days after injection), mice were randomized into two treatment groups of three mice each (two flank tumors in each mouse). The first (control) group received vehicle solution (DMSO diluted in 400 µl Saline-Glucose), and the second group received 30 mg/kg gefitinib in 400 µl DMSO and Saline-Glucose, both 5 days/week by i.p. injection for 3–4 weeks.

The effect of c-Cbl overexpression on tumor growth was assessed in mice with flank-injected HSC3/EGFR-GFP/Cbl or HSC3/EGFR-GFP/EV cells. When tumors were beginning to form (<50 mm$^3$; 8 days after injection), doxycycline (0.5 mg/ml) was added to the drinking water to induce c-Cbl expression, and tumor growth was monitored for 3–4 weeks.

In some experiments, 5 µg EGF-Rh in 100 µl PBS was injected in tail vein, and tumors were fixed or snap-frozen 1 hr after injection. In other experiments, HSC3/EGFR-GFP tumor-bearing mice were injected i. p. with gefitinib (30 mg/kg) or a mixture of Dyngo-4a and Pitstop2 (1.125 mM/each; 400 µl Saline-Glucose) 2 hr prior EGF-Rh injection.

## Intravital imaging of tumors

For intravital imaging, mice harboring HSC3/EGFR-GFP cell-derived tumors (150–250 mm$^3$) were anesthetized (Ketamine 100 mg/kg, Xylazine 12.5 mg/kg). Tumors were exposed by cutting the skin surrounding the tumor and holding the tumor steady with metal lifter to minimize motion artefacts caused by heart beat and breathing. To monitor EGF-Rh in tumors, 5 µg of EGF-Rh in 100 µl PBS was injected in the tail vein, and imaging was performed 1 hr after the injection. GenTeal eye gel (Novartis; Basel, Switzerland; refractive index 1.33) was placed on the tumor to interface with the two-photon objective. The eye gel has the same refractive index as water and is readily compatible with water immersion objectives. Time lapse image series were acquired using a Nikon multiphoton microscope (with 25x WI objective 1.15NA) at 1/4 frames per second with excitation at 800 nm and emission through 500–550 nm and 570–620 nm filters for GFP and rhodamine detection, respectively.

## Confocal imaging of cryosections and cultured cells in vitro

Tumor xenografts were dissected, fixed in 4% paraformaldehyde for 8 hr and processed into cryoblocks according to the standard procedure. Briefly, tumors were incubated in 30% sucrose for 16 hr, embedded in O.C.T compound (Fisher HealthCare, Coraopolis, PA) and stored at −80°C. Cryosections (10 µm) were cut on a cryostat (Leica CM1950; Leica, Wetzlar, Germany). Sections were washed with PBS and permeabilized with 0.1% Triton in PBS for 15 min for staining of nuclei with Hoechst33342. For immunolabeling, cryosections were permeabilized in 0.1%Triton X-100/2%BSA/ PBS for 20 min, incubated with the primary antibody (1:750, 2 hr at room temperature), washed and incubated with the secondary antibody labelled with Cy3 or Cy5 (1: 500, 1 hr at room temperature).

Cultured cells were grown on glass coverslips. Living or fixed cells were imaged as described (*Pinilla-Macua et al., 2016*).

Images from tumor sections or cultured cells were acquired using a spinning disk confocal imaging system based on a Zeiss Axio Observer Z1 inverted fluorescence microscope (with 63x Plan Apo PH NA 1.4) system, controlled by SlideBook6 software (Intelligent Imaging Innovation, Denver, CO) as described previously (*Pinilla-Macua et al., 2016*). The montages of multiple images were generated using SlideBook6. All image acquisition settings were identical for all experimental variants in each experiment.

## EGFR phosphorylation, ubiquitylation and signaling in cultured cells and tumors

For generation of the calibration graphs, cells were stimulated with EGF or TGFα for either 10 min (acute) or with EGF for 12 hr (the conditioned media was replaced by the same fresh medium after 4 and 8 hr). Cells were washed with ice-cold $Ca^{2+}$, $Mg^{2+}$-free PBS and lysed in TGH lysis buffer supplemented with protease and phosphatase inhibitors as described in *Pinilla-Macua et al., 2016*. Lysates were cleared by centrifugation and either electrophoresed (150 µg protein) or used for EGFR immunoprecipitation (1–1.5 mg protein) with the antibody 528 (10 µg/sample).

Snap-frozen tumor xenografts and tumor specimens from HNSCC patients were solubilized into RIPA buffer [20 mM Tris pH 7.5, 0.5% Igepal, 1% sodium deoxycholate, 1 mM sodium glycerophosphate, 2.5 mM Na pyrophosphate, 150 mM NaCl, 1 mM EDTA/EGTA, 1 mM orthovanadate,10 mM N-ethylmaleimide (NEM), protease inhibitors and PhosSTOP tablets (ROCHE Diagnostics, Manheim, Germany)] using Dounce homogenizer. Precleared lysates were either electrophoresed (350 µg), or used for EGFR immunoprecipitation (3.5 mg protein) with the antibody 528 (10 µg/sample).

The lysates and immunoprecipitates were resolved by SDS-PAGE (10%- or 7.5% gels, respectively), followed by the transfer to the nitrocellulose membrane. Western blotting was performed by incubating with appropriate primary antibodies followed by secondary antibodies conjugated to far-red fluorescent dyes (IRDye-680 and −800) and detection using an Odyssey Li-COR system. Quantifications were performed using Li-COR software. pY20 conjugated to horseradish peroxidase (HRP) was detected using enhanced chemiluminescence kit from Pierce (Rockford, IL, USA). Quantifications were performed using Image J 1.48 v software (*Schneider et al., 2012*).

## EGFR degradation

HSC3/EGFR-GFP/Cbl cells grown with or without doxycycline were incubated in DMEM supplemented with 0.1% BSA with or without EGF for 16 hr at 37°C. Cells were lysed in TGH in which NEM and sodium orthovanadate were omitted as described (*Huang et al., 2006*). Pre-cleared lysates (150 µg protein) were resolved by SDS-PAGE followed by the transfer to the nitrocellulose membrane. Western blotting and quantifications were performed as described above.

## HNSCC tumor specimens

Frozen and fresh HNSCC patient tumor specimens were acquired under the auspices of the SPORE in Head-and-Neck Cancer (HNC) (University of Pittsburgh). Fresh samples obtained immediately after surgery were snap-frozen in liquid nitrogen. HNSCC tumor specimens were homogenized and processed as described above for HSC3/EGFR-GFP tumor xenografts.

## Statistical data analysis

Statistical analysis was performed using GraphPad Prism version 7.00 for Windows, GraphPad Software, La Jolla, CA (www.graphpad.com). Unpaired t-test comparing two experimental groups and two-tailed p values were normally used except *Figure 8C*. Biological replicates were performed with $n \geq 3$.

## Acknowledgements

We are grateful to Drs. Sara Sigismund, Adam Kwiatkowski and Steffi Oesterreich for the gifts of reagents. We thank Dr. Roberto Weigert (National Cancer Institute) for the valuable advice and discussion of intravital imaging experiments, Dr. David Drubin, (University of California Berkeley) for generous help with gene-editing, plasmid sharing and advice; and Mr. Greg Gibson (Center for Biological Imaging, University of Pittsburgh) for technical help with the multiphoton microscope imaging. Supported by NIH/NCI grants CA089151 and GM124186 (AS), and UPCI HNC SPORE IO1-BX003456 (UD).

# Additional information

## Funding

| Funder | Grant reference number | Author |
|---|---|---|
| NIH Office of the Director | CA089151 | Itziar Pinilla-Macua<br>Alexander Sorkin |
| NIH Office of the Director | BX003456 | Umamaheswar Duvvuri |
| University of Pittsburgh Cancer Institute | UPCI HNC SPORE IO1-BX003456 | Umamaheswar Duvvuri |
| NIH Office of the Director | GM124186 | Alexander Sorkin |

The funders had no role in study design, data collection and interpretation, or the decision to submit the work for publication.

## Author contributions

Itziar Pinilla-Macua, Formal analysis, Validation, Investigation, Methodology, Writing—original draft; Alexandre Grassart, Investigation, Methodology; Umamaheswar Duvvuri, Resources; Simon C Watkins, Resources, Writing—review and editing; Alexander Sorkin, Conceptualization, Supervision, Funding acquisition, Visualization, Project administration, Writing—review and editing

## Author ORCIDs

Alexander Sorkin (iD) http://orcid.org/0000-0002-4446-1920

## Ethics

Human subjects: These study were exempt from the informed consent requirement (exemption 4). Specimens are collected under the auspices of our SPORE in Head and Neck Cancer. IRB approval date (6/13/2017).
Animal experimentation: This study was performed in strict accordance with the recommendations in the Guide for the Care and Use of Laboratory Animals of the National Institutes of Health. All of the animals were handled according to approved institutional animal care and use committee (IACUC) protocols (#16088888) of the University of Pittsburgh.

## Decision letter and Author response

Decision letter https://doi.org/10.7554/eLife.31993.019
Author response https://doi.org/10.7554/eLife.31993.020

# Additional files

## Supplementary files
• Transparent reporting form
DOI: https://doi.org/10.7554/eLife.31993.017

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
