## [Decision Letter]

Thank you for submitting your article "EGF receptor signaling, phosphorylation, ubiquitylation and endocytosis in tumors in vivo" for consideration by *eLife*. Your article has been reviewed favorably by three peer reviewers, and the evaluation has been overseen by a Reviewing Editor and John Kuriyan as the Senior Editor. The reviewers have opted to remain anonymous.

This is very nice study that looks at the activation and trafficking of EGFR in situ using a 3D tumor model. To accomplish this, the authors use gene editing techniques to tag endogenous EGFR with GFP in the human squamous cell carcinoma cell line HSC3. They demonstrate that EGFR-GFP appears to be normally activated and processed in the modified cells. HSC3/EGFR-GFP cells form tumors when injected into the tongues or flanks of mice, and the authors compare the behavior of EGFR-GFP in cell culture and in tumors using multiple methods including intravital fluorescence microscopy and confocal microscopy of fixed tumor sections. The authors show that the activation, ubiquitylation, and endocytosis of EGFR-GFP is similar in cells and tumors and use these studies to estimate effective high-affinity EGF ligand concentrations in mouse flank and tongue of ~0.2 and 0.6 ng/ml, respectively. The authors demonstrate that at these relatively low ligand concentrations the principal downstream pathway activated is the Erk pathway and that Cbl-mediated targeting to lysosomes is not strongly engaged.

Technically, the studies are extremely well done, well controlled and internally consistent. The data presented are generally clean. The fluorescent images are beautiful. The most novel aspect of this study is the careful comparison between the dynamics and activation of the tumor cells and their in vitro counterparts, thus providing a basis for extrapolating data from in vitro studies to observations of receptor dynamics and tumor responses in situ. The reviewers appreciate the fact that the authors take pains to carry out their experiments with apparently physiological levels of ligand and receptor as well as to compare in vitro and in vitro results. This is extraordinarily useful and thus this is potentially a very significant study.

The reviewers have identified several issues that must be addressed before the manuscript can be considered for publication. Note that it should be possible to address these issues without further experimentation, although any additional data that help clarify these points would be welcome.

1) It is the positioning of the paper as having made a "discovery" that tumors show low levels of occupied receptors that is its weakest aspect. Why would the authors have expected anything else (see specific citations given below)? Serum and plasma concentrations of EGF and other growth factors are low (<1ng/ml) and there are no obvious sources for high ligand concentrations. Most of the literature over the last two decades suggests that autocrine signaling is the primary determinant driving EGFR occupancy in cells. For example, mutations that abrogate the activity of ligand "sheddases" (e.g. TACE) have essentially the same effect as knocking out the EGFR, demonstrating the critical role of local production of ligands in regulating EGFR activity (PMID: 9812885). There is also a large literature on regulation of autocrine ligands which shows that (a) their rate of production controls the level of activation of the EGFR in situ, (b) they cannot be effectively measured in situ because they are essentially consumed as quickly as they are produced and (c) they are produced at extremely low levels (e.g. see PMID: 11493669). These prior studies support both the approach and conclusions of the authors, but observing low levels of receptor activation in vivo certainly should not have come as such a surprise.

The idea that high concentrations of EGFR ligands (e.g. >1ng/ml) are not physiologically relevant has been extensively discussed in the literature for many years and thus this conclusion is confirmatory at best. For example, mitogenesis (Albeck et al., 2013), endosomal sorting (Villasenor et al., 2015), adaptor phosphorylation (Krall et al., 2011), and ERK phosphorylation (Shi et al., 2016) all saturate at around 1ng/ml EGF, to cite just a handful of recent papers. It is true that studies continue to be published that treat cells with non-physiological levels of ligands, but this is usually because of insensitivity of analytical measurement techniques than a belief that high EGF levels recapitulate in vivo reality.

Please revise the manuscript so as to place the work in the most appropriate context, given prior findings. In this way, the important achievements of the work can be best appreciated.

2) The authors use a tumor type that overexpress EGFR. Thus, despite the use of endogenous labeling techniques, they are still dealing with cells overexpressing the EGFR. This should be more clearly pointed out as a caveat of their studies.

3) The calculation of the concentration of EGF ligands in the tumor is very indirect and we have doubts about the extent to which it should be considered reliable. And there is a nagging question about the extent to which the residual 50% of wild type EGF receptors in the gene-edited cells contribute to the signaling and internalization properties of the cells and tumors. The authors should address both points to the best of their ability in the revised manuscript.

---

## [Author Response]

The reviewers have identified several issues that must be addressed before the manuscript can be considered for publication. Note that it should be possible to address these issues without further experimentation, although any additional data that help clarify these points would be welcome.1) It is the positioning of the paper as having made a "discovery" that tumors show low levels of occupied receptors that is its weakest aspect. Why would the authors have expected anything else (see specific citations given below)? Serum and plasma concentrations of EGF and other growth factors are low (<1ng/ml) and there are no obvious sources for high ligand concentrations. Most of the literature over the last two decades suggests that autocrine signaling is the primary determinant driving EGFR occupancy in cells. For example, mutations that abrogate the activity of ligand "sheddases" (e.g. TACE) have essentially the same effect as knocking out the EGFR, demonstrating the critical role of local production of ligands in regulating EGFR activity (PMID: 9812885). There is also a large literature on regulation of autocrine ligands which shows that (a) their rate of production controls the level of activation of the EGFR in situ, (b) they cannot be effectively measured in situ because they are essentially consumed as quickly as they are produced and (c) they are produced at extremely low levels (e.g. see PMID: 11493669). These prior studies support both the approach and conclusions of the authors, but observing low levels of receptor activation in vivo certainly should not have come as such a surprise.The idea that high concentrations of EGFR ligands (e.g. >1ng/ml) are not physiologically relevant has been extensively discussed in the literature for many years and thus this conclusion is confirmatory at best. For example, mitogenesis (Albeck et al., 2013), endosomal sorting (Villasenor et al., 2015), adaptor phosphorylation (Krall et al., 2011), and ERK phosphorylation (Shi et al., 2016) all saturate at around 1ng/ml EGF, to cite just a handful of recent papers. It is true that studies continue to be published that treat cells with non-physiological levels of ligands, but this is usually because of insensitivity of analytical measurement techniques than a belief that high EGF levels recapitulate in vivo reality.Please revise the manuscript so as to place the work in the most appropriate context, given prior findings. In this way, the important achievements of the work can be best appreciated.

We certainly agree with the reviewers’ points that a number of studies suggested that (1) low EGFR ligand concentrations are physiological; (2) the ERK pathway is the most sensitive to the EGFR activity, and (3) shedding enzymes like TACE are the important determinants of the activity of EGFR ligands like TGFa in vivo. Thus, indeed, low levels of EGFR activity in vivo could have been expected, perhaps, except for the liver. That said, it is important to stress that studies listed above and most of other quantitative studies of EGFR signaling were performed in cultured cells or normal tissues, whereas our studies are aimed at the quantitative analysis of EGFR activities in tumors in vivo. Several major groups maintain that they study EGFR using high ligand concentrations because an autocrine production of ligands may be elevated in tumors in vivo (due to increased gene expression, trans-activation of shedding enzymes by other signaling systems, and other mechanisms). Additionally, tumor cells may be exposed to a high concentration of EGFR ligands due to increased paracrine ligand production, e.g. by infiltrating TAMs (tumor-associated macrophages). Clearly, the regulation of ligand levels is tumor-type-specific, and these levels cannot be simply extrapolated from in vitro studies, but must be directly examined in a particular experimental model of tumorigenesis. Therefore, it was essential to perform studies of EGFR activities directly in EGFR-dependent tumors in vivo, and using comparative analysis of in vitro versus in vivo EGFR activities in the same cells.

I must admit that extremely low levels of EGFR endocytosis and activity that we observed in tumors in vivo were surprising to me, despite my own views on this issue and my efforts over the years to convince the EGFR community that low EGFR ligand concentrations should be used in experiments. Importantly, our experiments showed that even with the cell-to-cell heterogeneity of the EGFR activity in the tumor, the number of active EGFR per tumor cell in vivo would not exceed the threshold leading to the activation of a multitude of signaling pathways and clathrin-independent internalization of EGFR observed in vitro when receptor-saturating ligand concentrations are used.

We greatly appreciate that the reviewers recognize the importance of EGFR ligand concentrations and share our view that low EGFR ligand concentrations are physiologically relevant. Unfortunately, this is not the case for the vast majority of EGFR researchers. For example, among most recent 15 Pubmed citations on the keyword “EGFR”, 13 papers used EGF at concentrations 20 ng/ml or higher, the most popular EGF concentration was 100 ng/ml, and none used ligands at less than 1 ng/ml. In most of these papers, measurements of EGF response were routine, and high concentrations were used not “because of insensitivity of analytical measurement techniques”. In the revised manuscript, we emphasized that qualitatively different conclusions can be made about the biology of the EGFR system using high versus low ligand concentrations (Discussion section). This point can be further illustrated by an example of a recent high-profile publication (Caldieri et al., 2017) describing a novel mechanism of clathrin-independent endocytosis of EGFR. This is an excellent and pioneering study, which may, however, mislead EGFR researchers who are studying the role of EGFR endocytosis in tumorigenesis and developing approaches for altering receptor trafficking in tumors.

In summary, we made following changes in the revised manuscript:

1) We cite and discuss papers listed above by the reviewers (Introduction; Discussion section) and additionally, with regards to the saturation of the endosomal sorting, we cite an earlier study by French et al. that originally demonstrated the saturation of EGF degradation system by increasing ligand concentrations (Introduction).

2) We state in subsection “ERK1/2 pathway is the primary signaling axis that is significantly activated by picomolar EGFR ligand concentrations” that the pattern of activation of various signaling pathways by EGFR in HSC3 cells in vitro is similar to that pattern observed in other cell types in previous studies.

3) We cite additional papers that propose increased autocrine and paracrine production of EGFR ligands that are accessible to cells in tumors (Discussion section) to emphasize the importance of our findings obtained by the direct examination of the activity and endocytic mechanisms of EGFR in tumors in vivo.

4) The expression “surprisingly” was omitted in the Introduction.

2) The authors use a tumor type that overexpress EGFR. Thus, despite the use of endogenous labeling techniques, they are still dealing with cells overexpressing the EGFR. This should be more clearly pointed out as a caveat of their studies.

We have chosen HSC3 cells because they (i) were derived from human patient oral carcinoma; (ii) resemble receptor levels of human HNSCC, 90% of which overexpress wild-type EGFR, and (iii) recapitulate the growth dependence of tumors on wild-type EGFR. We clarified the issue of the receptor levels and our rationale for choosing HSC3 in subsection “Generation and characterization of gene-edited HSC3 cells expressing endogenous GFP-tagged EGFR”.

3) The calculation of the concentration of EGF ligands in the tumor is very indirect and we have doubts about the extent to which it should be considered reliable. And there is a nagging question about the extent to which the residual 50% of wild type EGF receptors in the gene-edited cells contribute to the signaling and internalization properties of the cells and tumors. The authors should address both points to the best of their ability in the revised manuscript.

Though we have made all possible efforts to perform quantitative in vitro:in vivo analysis in each set of experiments and conducted double-blind counting of tumor cells accessible to systemic EGF-Rh, we emphasize in the manuscript the Discussion section that our measurements predict a range of the ligand concentration rather than provide a precise measurement of these concentrations.

With regards to the presence of both untagged and GFP-tagged EGFR in HSC3/EGFR-GFP cells, our biochemical measurements showed no evidence for different signaling (Figure 1) and trafficking properties of these two EGFR species. While we cannot directly measure endocytosis of untagged EGFR in HSC3/EGFR-GFP cells, Figure 8—figure supplement 1 demonstrates an essentially similar extent of EGF-induced degradation of both these EGFR species. To strengthen this point, we performed new experiments in which total EGFR was immunolabeled on tumor sections (new Figure 2—figure supplement 1). These experiments demonstrate high extent of co-localization and essentially similar localization patterns of GFP and total EGFR, which is indicative of no additional endocytic activity by untagged EGFR. See subsection “EGFR-GFP localization and trafficking in HSC3/EGFR-GFP tumor xenografts”.